# Identifiability Matters: Revealing the Hidden Recoverable Condition in Unbiased Learning to Rank

## Abstract

The application of Unbiased Learning to Rank (ULTR) is widespread in modern systems for training unbiased ranking models from biased click logs. The key is to explicitly model a generation process for user behavior and fit click data based on examination hypothesis. Previous research found empirically that the true latent relevance can be recovered in most cases as long as the clicks are perfectly fitted. However, we demonstrate that this is not always achievable, resulting in a significant reduction in ranking performance. In this work, we aim to answer if or when the true relevance can be recovered from click data, which is a foundation issue for ULTR field. We first define a ranking model as identifiable if it can recover the true relevance up to a scaling transformation, which is enough for pairwise ranking objective. Then we explore an equivalent condition for identifiability that can be novely expressed as a graph connectivity test problem: if and only if a graph (namely identifiability graph, or IG) constructed on the underlying structure of the dataset is connected, we can guarantee that the relevance can be correctly recovered. When the IG is not connected, there may be bad cases leading to poor ranking performance. To address this issue, we propose two methods, namely node intervention and node merging, to modify the dataset and restore connectivity of the IG. Empirical results obtained on a simulation dataset and two LTR benchmark datasets confirm the validity of our proposed theorems and show the effectiveness of our methods in mitigating data bias when the relevance model is unidentifiable.

## 1 Introduction

The utilization of click data for Learning to Rank (LTR) methodologies has become prevalent in contemporary information retrieval systems, as the accumulated feedback effectively demonstrates the value of individual documents to users (Agarwal et al., 2019a) and is comparatively effortless to amass on an extensive scale. Nevertheless, inherent biases stemming from user behaviors are presented within these datasets (Joachims et al., 2007). One example is position bias (Joachims et al., 2005), wherein users exhibit a propensity to examine documents situated higher in the rankings, causing clicks to be biased with the position. Removing these biases is critical since they hinder learning the correct ranking model from click data. To address this issue, Unbiased Learning to Rank (ULTR) is developed to mitigate these biases (Joachims et al., 2017). The central idea is to explicitly model a generation process for user clicks using the **examination hypothesis**, which assumes that each document possesses a probability of being observed (depending on certain bias factors, *e.g.*, position (Joachims et al., 2017) or context (Fang et al., 2019)), and subsequently clicked based on its relevance to the query (depending on ranking features encoding the query and document), formulated as follows:

$$P(\text{click}) = \underbrace{P(\text{observation} \mid \text{bias factors})}_{\text{observation model}} \cdot \underbrace{P(\text{relevance} \mid \text{ranking features})}_{\text{ranking model}}.$$

In practice, a joint optimization process is employed to optimize the observation and ranking models in the form of the examination hypothesis (Wang et al., 2018; Ai et al., 2018a; Zhao et al., 2019; Guo et al., 2019), which is able to fit the click data efficiently.

However, rather than predicting the user's click probability, we're more interested in recovering the true relevance by the ranking model, which is the primary objective of the ULTR framework. Previous studies have found empirically that this can be achieved in most cases if the click probability is perfectly fitted (Ai et al., 2018a; Wang et al., 2018; Ai et al., 2021). Unfortunately, no existing literature has provided a guarantee to support this statement, and it is entirely possible that the relevance becomes *unrecoverable* in some cases. This has be verified in Oosterhuis (2022), which constructed a simplified dataset and showed that a perfect click model trained on this dataset can yield inaccurate and inconsistent relevance estimates. Despite the limited scale of their example, we contend that in real scenarios with large-scale data, the unrecoverable phenomenon can persist if there are excessive bias factors. We demonstrate it in the following example.

**Example 1.** *Consider a large-scale dataset where each displayed query-document pair comes with a **__distinct__** bias factor[1]. In this case, unfortunately, it becomes challenging to differentiate whether the ranking feature or the bias factor is influencing relevance, known as coupling effect (Chen et al., 2022a). One might train a naive ranker that predicts a constant one, paired with an observation model that maps the unique bias factor to the true click probability. The product of the two models is an accurate click estimation, but the estimated relevance is erroneous.*

While one of the current research trend is to incorporate more bias factors into datasets (Vardasbi et al., 2020a; Chen et al., 2021; 2022a; 2023) to accurately estimate observation probabilities, we argue that there is an urgent need to answer *if or when the relevance can be recovered*, which would establish a basis for the ULTR field. However, this problem is rather challenging since it heavily depends on the exact data collection procedure (Oosterhuis, 2022), and to the best of our knowledge, previous research has not yet provided a feasible solution to determine whether a ULTR dataset supports recovering the relevance.

In this work, we introduce a novel identifiability framework to address this core problem. Considering the ranking objective, we define that a ranking model is ***identifiable*** if *it can recover the true relevance probability up to a scaling transformation* in § 3. Identifiability is a sufficient condition for correct ranking. We prove that the identifiability of a ranking model depends on an underlying structure of the dataset, which can be converted to a practical graph connectivity test problem: if and only if a graph (named as *identifiability graph*, or IG) constructed on the dataset is connected, we can guarantee the identifiability of the ranking model. In cases where the IG is disconnected, there may exist situations in which ranking fails. Through theoretical analysis and empirical verification, we observe that the probability of identifiability depends on the size of the dataset and bias factors. In particular, smaller datasets or a larger number of bias factors lead to a higher probability of IG disconnection. Besides, we observe that a real-world dataset, TianGong-ST (Chen et al., 2019), is unidentifiable due to the presence of numerous bias factors. These findings offer important guidance for the future design of ULTR datasets and algorithms.

Based on the theory, we further explore strategies for addressing datasets characterized by a disconnected IG. We propose two methods to enhance IG connectivity and ensure the identifiability of ranking models: (1) node intervention (§ 4.1), which swaps documents between two bias factors (mainly positions) to augment the dataset. While the intervention is common in ULTR literature (Joachims et al., 2017), our proposed approach is able to explore the minimal required number of interventions and significantly reduces the cost; and (2) node merging (§ 4.2), which merges two bias factors together and assume identical observation probabilities. We conducted extensive experiments using a fully simulated dataset and two real LTR datasets to validate our theorems and demonstrate the efficacy of our methods when dealing with disconnected IGs.

To the best of our knowledge, we are the first to study the identifiability of ranking models in the current ULTR framework. The main contributions of this work are:

1. We propose the concept of identifiability which ensures the capacity to recover the true relevance from click data, and convert it into a graph connectivity test problem from the dataset perspective.

2. We propose methods to handle the unidentifiable cases by modifying datasets.

3. We give theoretical guarantees and empirical studies to verify our proposed frameworks.

---

[1]This is feasible if we incorporate enough fine-grained bias factors discovered by previous studies, such as positions, other document clicks (Vardasbi et al., 2020a; Chen et al., 2021), contextual information (Fang et al., 2019), representation styles (Liu et al., 2015; Zheng et al., 2019) or a variety of other contextual features (Ieong et al., 2012; Sun et al., 2020; Chen et al., 2020; Sarvi et al., 2023).

## 2 PRELIMINARIES

Given a query $q \in \mathcal{Q}$, the goal of learning to rank is to learn a ranking model to sort a set of documents. Let $\boldsymbol{x} \in \mathcal{X}$ denote the query-document ranking features, and the ranking model aims to estimate a relevance score with $\boldsymbol{x}$. Since the ground truth relevance is difficult to obtain at scale, in practice users' click logs are often used as labels to train the model. While user clicks are usually biased from the true relevance, researchers propose examination hypothesis to factorize the biased clicks into relevance probability and observation probability, which can be formulated as:

$$c(\boldsymbol{x}, \boldsymbol{t}) = r(\boldsymbol{x}) \cdot o(\boldsymbol{t}), \tag{1}$$

where $c(\boldsymbol{x}, \boldsymbol{t}) \in [0, 1]$, $r(\boldsymbol{x}) \in [0, 1]$ and $o(\boldsymbol{t}) \in [0, 1]$ denote the probability that the document is clicked, relevant and observed, respectively. $\boldsymbol{t} \in \mathcal{T}$ denotes bias factors that cause clicks to be biased, such as position (Joachims et al., 2017), other document clicks (Vardasbi et al., 2020a; Chen et al., 2021), contextual information (Fang et al., 2019) or the representation style (Liu et al., 2015). Let $\mathcal{D} = \{(\boldsymbol{x}_i, \boldsymbol{t}_i)\}_{i=1}^{|\mathcal{D}|}$ denote pairs of ranking features and bias factors. To simplify the analysis, we suppose all bias factors $\boldsymbol{t} \in \mathcal{T}$ and features $\boldsymbol{x} \in \mathcal{X}$ appear in $\mathcal{D}$. By explicitly modeling the bias effect via observation, it is expected to attain an unbiased estimate of the ranking objective.

To jointly obtain relevance score and observation score, we optimize a ranking model $r'(\cdot)$ and an observation model $o'(\cdot)$ to fit clicks, which can be formulated as:

$$\mathcal{L} = \sum_{i=1}^{|\mathcal{D}|} l(r'(\boldsymbol{x}_i) \cdot o'(\boldsymbol{t}_i), c_i), \tag{2}$$

where $c_i$ denotes the click, and $l(\cdot, \cdot)$ denotes a loss function of interest, such as mean square error or cross entropy error.

## 3 IDENTIFIABILITY

Most current work presumes that optimizing Eq.(2) can obtain correct ranking model (Wang et al., 2018; Chen et al., 2022a; 2023). Unfortunately, there are no guarantees about it: all we know is that the product of two model outputs is correct, but sometimes they lead to a poor ranking performance as shown in the Example 1. Our goal is to explore a fundamental condition that the underlying latent relevance function can be recovered (*i.e.*, **identifiable**), formulated as:

$$r(\boldsymbol{x}) \cdot o(\boldsymbol{t}) = r'(\boldsymbol{x}) \cdot o'(\boldsymbol{t}) \quad \Longrightarrow \quad r = r'.$$

Note that it's intractable to directly recover the exact relevance model since we can scale $r(\cdot)$ by $n$ times and scale $o(\cdot)$ by $1/n$ times while keeping their product remaining the same. In practice, we are often interested in relevance that is identifiable up to a scaling transformation, which is enough for pairwise ranking objective. Thus, we introduce the following definition for identifiability:

**Definition 1** (Identifiable). *We say that the relevance model is identifiable, if:*

$$r(\boldsymbol{x}) \cdot o(\boldsymbol{t}) = r'(\boldsymbol{x}) \cdot o'(\boldsymbol{t}), \forall (\boldsymbol{x}, \boldsymbol{t}) \in \mathcal{D} \quad \Longrightarrow \quad \exists C > 0, \text{ s.t. } r(\boldsymbol{x}) = Cr'(\boldsymbol{x}), \forall \boldsymbol{x} \in \mathcal{X}.$$

The identifiability is a sufficient condition for ranking guarantees. If there are no guarantees on the identifiability, there may be a bad case like Example 1. Our main result in the following theorem (the proof is delegated to Appendix B.1) shows that the identifiability is related to the underlying structure of the dataset, which can be easily mined.

**Theorem 1** (Identifiability condition). *The relevance model is identifiable, **if and only if** an undirected graph $G = (V, E)$ is connected, where $V$ is a node set and $E$ is an edge set, defined as:*

$$V = \{v_1, v_2, \cdots, v_{|\mathcal{T}|}\},$$
$$E = \{(v_s, v_t) \mid \exists \boldsymbol{x} \in \mathcal{X}, \text{ s.t. } (\boldsymbol{x}, \boldsymbol{s}) \in \mathcal{D} \wedge (\boldsymbol{x}, \boldsymbol{t}) \in \mathcal{D}\},$$

*We refer to this graph as **identifiability graph (IG)**.*

**Remark 1.** *The relevance identifiability is equivalent to a graph connectivity test problem. The IG is constructed as follows: we first create nodes for each bias factors $t \in \mathcal{T}$. If there exists a feature appearing with two bias factors together, add an edge between the two nodes. Theorem 1 establishes connections to recent ULTR research (Agarwal et al., 2019c; Oosterhuis, 2022; Zhang et al., 2023), which are elaborated in § 6.*

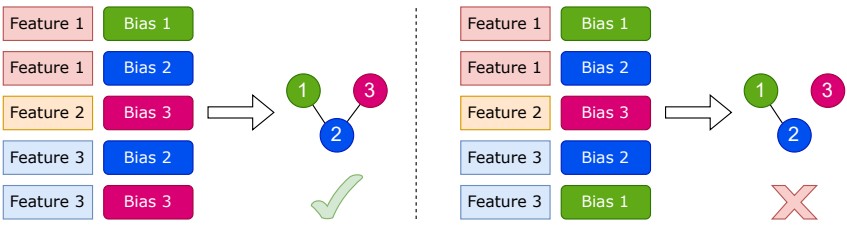

Figure 1: Examples for identifiable case and unidentifiable case.

Figure 1 illustrates examples for applying Theorem 1 to verify identifiability of the datasets. In the left figure, bias factors 1 and 2 are connected through feature 1, and bias factors 2 and 3 are connected through feature 3. As a result, the graph is connected and the relevance is identifiable. Conversely, the right figure depicts a scenario where bias factor 3 remains isolated, leading to unidentifiability of relevance. Based on Theorem 1, we illustrate the identifiability check algorithm in Appendix D.1.

Next, we are interested in finding out the likelihood of a ranking model being identifiable. Particularly, we aim to estimate the varying impact of dataset size $|\mathcal{D}|$, the number of features $|\mathcal{X}|$, and the number of bias factors $|\mathcal{T}|$ on the probability of identifiability to provide an intuitive understanding. Nevertheless, precise calculation of this probability is challenging due to the unknown nature of the generation process for $\mathcal{D}$. To address this, the following corollary assumes a simplified distribution for $\mathcal{D}$ and provides an estimate. We defer the proof to Appendix B.2.

**Corollary 1** (Estimation of identifiability probability)**.** *Considering the following simplified case[2]: each feature $x \in \mathcal{X}$ and bias factor $t \in \mathcal{T}$ are selected independently and uniformly to construct a dataset $\mathcal{D}$. Then the probability of identifiability can be estimated by:*

$$P\left(\text{identifiability} \mid |\mathcal{D}|, |\mathcal{X}|, |\mathcal{T}|\right) \sim 1 - |\mathcal{T}| \exp\left(-|\mathcal{D}| + f\right),$$

$$\text{where } f = |\mathcal{X}||\mathcal{T}| \log \left[2 - \exp\left(-\frac{|\mathcal{D}|}{|\mathcal{X}||\mathcal{T}|}\right)\right].$$

**Remark 2.** *Note that $\lim_{|\mathcal{T}| \to +\infty} f = |\mathcal{D}|$. This implies that when $|\mathcal{T}|$ is sufficiently large, the probability of identifiability decays linearly, which requires a sufficiently large dataset $|\mathcal{D}|$ to offset this effect. Given that one of the research trend in ULTR is to digest more bias factors while maintaining a constant dataset size (Chen et al., 2021; 2023), this conclusion is important for future research.*

## 4 DEALING WITH UNIDENTIFIABLE DATASET

In this section, we discuss how to deal with datasets that the relevance is unidentifiable. We propose two methods applied on datasets, namely *node intervention* and *node merging*, to establish connectivity within the IG. The former method necessitates the inclusion of supplementary data which enables the accurate recovery of relevance, while the latter method only needs existing data but may introduce approximation errors. We provide an illustration for these methods in Appendix C. In practice, the selection of the appropriate method is contingent upon the specific problem requirements.

### 4.1 NODE INTERVENTION

Given that unidentifiability results from the incomplete dataset, one method is to augment datasets by swapping some pairs of documents between different bias factors (mainly positions). Swapping is a widely adopted technique that can yield accurate observation probabilities (Joachims et al., 2017). However, existing methods are often heuristic in nature, lacking a systematic theory to guide the swapping process, and sometimes performing unnecessary swapping operations. Instead, with the help of IGs, we can find the minimum number of critical documents that require swapping.

Our basic idea is that (1) find the connected components in the IG; (2) for any two components, find a node in each of them; and (3) for these two nodes (representing two bias factors, *e.g.*, $t_1$ and $t_2$), find a feature (denoted by $x$) related to one of them (denoted by $t_1$), and swap it to another bias factor

---

[2]As a supplement, we also give an empirical analysis in § 5.2 in a more realistic and complicated case.

(denoted by $\boldsymbol{t}_2$). This operation will generate a new data point $(\boldsymbol{x}, \boldsymbol{t}_2)$. Since $(\boldsymbol{x}, \boldsymbol{t}_1)$ has already been in the dataset, this process will connect $\boldsymbol{t}_1$ and $\boldsymbol{t}_2$ in the IG and thus connect two components. Repeat this process until the IG is connected. We refer to this method as **node intervention**.

However, there are too many choices that the features and bias factors can be selected. How to design appropriate selection criteria is important since we aim to make the fewest interventions. Note that the observation probability of some bias factors is relatively low (*e.g.*, the last position in the list), necessitating the collection of more clicks to obtain a valid click rate. These bias factors are not suitable for swapping. To account for the number of clicks, instead of simply assuming that we observe an accurate click probability $r(\boldsymbol{x}) \cdot o(\boldsymbol{t})$, here we assume that we can only observe a random variable for click rate which is an average of $N$ clicks: $1/N \sum_{i=1}^{N} c_i$. $c_i \in \{0, 1\}$ is a binary random variable sampling from a probability $r(\boldsymbol{x}) \cdot o(\boldsymbol{t})$, indicating whether the query-document pair is clicked. Definition 1 can be seen as a special case when $N \to +\infty$. Based on it, we establish the following proposition, with its proof delegated to Appendix B.3.

**Proposition 1.** *For a feature $\boldsymbol{x}$ and two bias factors $\boldsymbol{t}_1, \boldsymbol{t}_2$, suppose $r'(\boldsymbol{x}) \cdot o'(\boldsymbol{t}) = 1/N \sum_{i=1}^{N} c_i(\boldsymbol{x}, \boldsymbol{t})$, where $\boldsymbol{t} \in \{\boldsymbol{t}_1, \boldsymbol{t}_2\}$ and $c_i(\boldsymbol{x}, \boldsymbol{t})$ are random variables i.i.d. sampled from Bernoulli$(r(\boldsymbol{x}) \cdot o(\boldsymbol{t}))$ for $1 \le i \le N$. Assuming $r(\boldsymbol{x})$, $o(\boldsymbol{t}_1)$ and $o(\boldsymbol{t}_2)$ are non-zero, then:*

$$\mathbb{E}\left[\frac{o'(\boldsymbol{t}_1)}{o(\boldsymbol{t}_1)} - \frac{o'(\boldsymbol{t}_2)}{o(\boldsymbol{t}_2)}\middle| r'(\boldsymbol{x})\right] = 0,$$

$$\mathbb{V}\left[\frac{o'(\boldsymbol{t}_1)}{o(\boldsymbol{t}_1)} - \frac{o'(\boldsymbol{t}_2)}{o(\boldsymbol{t}_2)}\middle| r'(\boldsymbol{x})\right] = \frac{1}{NR}\left[\frac{1}{r(\boldsymbol{x}) \cdot o(\boldsymbol{t}_1)} + \frac{1}{r(\boldsymbol{x}) \cdot o(\boldsymbol{t}_2)} - 2\right],$$

*where $R = r'(\boldsymbol{x})^2/r(\boldsymbol{x})^2$.*

**Remark 3.** *As $N$ or $r(\boldsymbol{x})o(\boldsymbol{t})$ increases, the variance $\mathbb{V}$ decreases, leading to $o'(\boldsymbol{t})/o(\boldsymbol{t})$ approaching a constant zero. Therefore, the ranking model is identifiable according to Definition 1. In practice, optimal feature and bias factors can be chosen to minimize the variance and reduce the required $N$.*

Based on Proposition 1, for two connected components $G_A = (V_A, E_A)$ and $G_B = (V_B, E_B)$, we use the following process to connect $G_A$ and $G_B$, by minimizing $\mathbb{V}$ to facilitate identifiability:

$$\text{cost}(\boldsymbol{x}, \boldsymbol{t}_1, \boldsymbol{t}_2) = \frac{1}{r(\boldsymbol{x}) \cdot o(\boldsymbol{t}_1)} + \frac{1}{r(\boldsymbol{x}) \cdot o(\boldsymbol{t}_2)} - 2, \tag{3}$$

$$\boldsymbol{t}_1^{(A)}, \boldsymbol{t}_2^{(A)}, \boldsymbol{x}^{(A)} \leftarrow \arg\min_{\boldsymbol{t}_A \in V_A, \boldsymbol{t}_B \in V_B, \boldsymbol{x} \in \{\boldsymbol{x}_i | (\boldsymbol{x}_i, \boldsymbol{t}_A) \in \mathcal{D}\}} \text{cost}(\boldsymbol{x}, \boldsymbol{t}_A, \boldsymbol{t}_B), \tag{4}$$

$$\boldsymbol{t}_1^{(B)}, \boldsymbol{t}_2^{(B)}, \boldsymbol{x}^{(B)} \leftarrow \arg\min_{\boldsymbol{t}_A \in V_A, \boldsymbol{t}_B \in V_B, \boldsymbol{x} \in \{\boldsymbol{x}_i | (\boldsymbol{x}_i, \boldsymbol{t}_B) \in \mathcal{D}\}} \text{cost}(\boldsymbol{x}, \boldsymbol{t}_A, \boldsymbol{t}_B), \tag{5}$$

$$\boldsymbol{x}^{(A,B)}, \boldsymbol{t}^{(A,B)} \leftarrow \begin{cases} \boldsymbol{x}^{(A)}, \boldsymbol{t}_2^{(A)} & \text{if } \text{cost}(\boldsymbol{x}^{(A)}, \boldsymbol{t}_1^{(A)}, \boldsymbol{t}_2^{(A)}) \le \text{cost}(\boldsymbol{x}^{(B)}, \boldsymbol{t}_1^{(B)}, \boldsymbol{t}_2^{(B)}), \\ \boldsymbol{x}^{(B)}, \boldsymbol{t}_1^{(B)} & \text{otherwise.} \end{cases} \tag{6}$$

Here Eq.(3) defines a cost[3] of swapping $\boldsymbol{x}$ from $\boldsymbol{t}_1$ to $\boldsymbol{t}_2$ (or from $\boldsymbol{t}_2$ to $\boldsymbol{t}_1$) based on $\mathbb{V}$ derived by Proposition 1. We ignore $R$ since it is a constant when the relevance model is identifiable. Eq.(4) defines the process that we find a feature $\boldsymbol{x}^{(A)}$ related to a bias factor $\boldsymbol{t}_1^{(A)}$ (belongs to $G_A$) and swap it to another bias factor $\boldsymbol{t}_2^{(B)}$ (belongs to $G_B$). Reversely, Eq.(5) defines the process to swap the feature from $G_B$ to $G_A$. The final decision depends on the process with less cost (Eq.(6)). We refer to this cost as the intervention cost between $G_A$ and $G_B$. Finally, we add $(\boldsymbol{x}^{(A,B)}, \boldsymbol{t}^{(A,B)})$ to the dataset $\mathcal{D}$ and collect enough user clicks about it, which connects $G_A$ and $G_B$ together.

In the above process, two components are connected. Next, we show how to connect the entire IG. Consider an IG graph, denoted by $G$, which is composed of $K$ connected components: $G = G_1 \cup \cdots \cup G_K$. For these $K$ components, we construct another complete graph consisting of $K$ nodes, where the edge weight between the $i$-th and the $j$-th node is equivalent to the intervention cost between $G_i$ and $G_j$. A minimum spanning tree (MST) algorithm is then implemented on this complete graph to find edges with the minimal total weights required for connection. The intervention operation (Eq.(3) - Eq.(6)) is then executed on these edges. The comprehensive algorithm for node intervention is presented in Appendix D.2.

---

[3]The value of $r$ and $o$ in Eq.(3) can be any rational guess. For example, we can select a ranking model directly trained with biased clicks for $r$, and select a manually designed observation model for $o$. We can also use the node merging method, which we will discuss soon, to derive the initial guess for $r$ and $o$.

Compared to traditional intervention strategies that usually require random swapping for all queries (Joachims et al., 2017; Radlinski & Joachims, 2006; Carterette & Chandar, 2018; Yue et al., 2010), node intervention involves performing a mere $K - 1$ swaps when the IG consists of $K$ connected components. It should be noted that $K$ is typically smaller than the number of positions (assuming that positions are the sole bias factors), which is significantly less than the total number of queries. Thus, the utilization of node intervention leads to a substantial reduction in the count of online interventions, resulting in an enhanced user experience.

## 4.2 NODE MERGING

Despite node intervention being effective in achieving identifiability, it still requires additional online experiments, which can be time-consuming and may pose a risk of impeding user experience by displaying irrelevant documents at the top of the ranking list. What's worse, some types of bias factors may not be appropriate for swapping (*e.g.*, contextual information or other documents' clicks). Therefore, we propose another simple and general methodology for addressing the unidentifiability issue, which involves merging nodes from different connected components and assuming they have the same observation probability. We refer to this strategy as **node merging**.

Similar to node intervention, there are numerous options for selecting node pairs to merge. Note that merging two dissimilar nodes with distinct observation probabilities will inevitably introduce approximation errors, as stated in the following proposition (We defer the proof to Appendix B.4):

**Proposition 2** (Error bound of merging two components). *Suppose an IG $G = (V, E)$ consists of two connected components $G_1 = (V_1, E_1)$ and $G_2 = (V_2, E_2)$. If we merge two nodes $v_1 \in G_1$ and $v_2 \in G_2$, i.e., forcing $o'(\boldsymbol{t}') = o'(\boldsymbol{t}'')$ where $v_1$ and $v_2$ represent bias factors $\boldsymbol{t}'$ and $\boldsymbol{t}''$, then:*

$$r(\boldsymbol{x}) \cdot o(\boldsymbol{t}) = r'(\boldsymbol{x}) \cdot o'(\boldsymbol{t}) \quad \Longrightarrow \quad \left| \frac{r'(\boldsymbol{x}_1)}{r(\boldsymbol{x}_1)} - \frac{r'(\boldsymbol{x}_2)}{r(\boldsymbol{x}_2)} \right| \leq \left| \frac{o(\boldsymbol{t}') - o(\boldsymbol{t}'')}{o'(\boldsymbol{t}')} \right|, \forall \boldsymbol{x}_1, \boldsymbol{x}_2 \in \mathcal{X},$$

*where we suppose $r, r', o$ and $o'$ are not zero.*

**Remark 4.** *When $o(\boldsymbol{t}') = o(\boldsymbol{t}'')$, the relevance model is identifiable. Otherwise, if the gap between $o(\boldsymbol{t}')$ and $o(\boldsymbol{t}'')$ is large, $r'(\boldsymbol{x})/r(\boldsymbol{x})$ will change greatly, hurting the ranking performance.*

Therefore, we propose to merge similar nodes exhibiting minimal differences in their observation probabilities. We assume that each bias factor $\boldsymbol{t}$ can be represented using an $F$-dimensional feature vector $\boldsymbol{X_t}$ (namely bias features). We further assume that the more similar the vectors, the closer the corresponding observation probabilities. For instance, if positions are the only bias factors, we can consider the document rank as a 1-dimensional bias feature. It is reasonable that documents with similar positions will have similar observation probabilities.

Based on it, for two connected components $G_A = (V_A, E_A)$ and $G_B = (V_B, E_B)$, we can use the following process to connect $G_A$ and $G_B$:

$$\text{cost}(\boldsymbol{t}_1, \boldsymbol{t}_2) = ||\boldsymbol{X}_{\boldsymbol{t}_1} - \boldsymbol{X}_{\boldsymbol{t}_2}||, \tag{7}$$

$$\boldsymbol{t}_A^*, \boldsymbol{t}_B^* \leftarrow \arg\min_{t_A \in V_A, t_B \in V_B} \text{cost}(\boldsymbol{t}_A, \boldsymbol{t}_B). \tag{8}$$

Here Eq.(7) defines the merging cost to merge $\boldsymbol{t}_1$ and $\boldsymbol{t}_2$. Eq.(8) finds two bias factors $\boldsymbol{t}_A^*$ and $\boldsymbol{t}_B^*$ from two components that have the minimal merging cost. We refer to this cost as the merging cost between $G_A$ and $G_B$. Similar to node intervention, we use the MST algorithm to make the IG connect. The algorithm is almost the same as in § 4.1, while the only difference is that the edge weight of the complete graph is defined as the merging cost. We provide the full algorithm for node intervention in Appendix D.3. Besides, we provide an error bound for the node merging algorithm in Appendix B.5, showing that the error is bounded by the longest path in the MST.

Compared to node intervention, node merging performs on the offline dataset, making it a simple and time-efficient approach. However, merging bias factors brings additional approximation error which has the potential to adversely impact the ranking performance.

Table 1: Performance of different methods on the $K = 2$ simulation dataset under PBM bias (number of clicks = $10^6$). We report the average results as well as the standard deviations.

| | MCC | nDCG@1 | nDCG@3 | nDCG@5 | nDCG@10 |
|---|---|---|---|---|---|
| No debias | $0.641_{\pm.000}$ | $0.769_{\pm.000}$ | $0.713_{\pm.000}$ | $0.729_{\pm.000}$ | $0.858_{\pm.000}$ |
| DLA | $0.738_{\pm.101}$ | $0.838_{\pm.038}$ | $0.779_{\pm.070}$ | $0.820_{\pm.057}$ | $0.900_{\pm.031}$ |
| + Node intervention | $\mathbf{1.000}_{\pm.000}$ | $\mathbf{1.000}_{\pm.000}$ | $\mathbf{1.000}_{\pm.000}$ | $\mathbf{1.000}_{\pm.000}$ | $\mathbf{1.000}_{\pm.000}$ |
| + Node merging | $0.987_{\pm.000}$ | $\mathbf{1.000}_{\pm.000}$ | $\mathbf{1.000}_{\pm.000}$ | $\mathbf{1.000}_{\pm.000}$ | $\mathbf{1.000}_{\pm.000}$ |
| Regression-EM | $0.634_{\pm.095}$ | $0.805_{\pm.028}$ | $0.703_{\pm.033}$ | $0.760_{\pm.017}$ | $0.868_{\pm.015}$ |
| + Node intervention | $\mathbf{0.982}_{\pm.014}$ | $\mathbf{1.000}_{\pm.000}$ | $0.999_{\pm.002}$ | $0.998_{\pm.007}$ | $0.999_{\pm.002}$ |
| + Node merging | $0.975_{\pm.000}$ | $\mathbf{1.000}_{\pm.000}$ | $\mathbf{1.000}_{\pm.000}$ | $\mathbf{1.000}_{\pm.000}$ | $\mathbf{1.000}_{\pm.000}$ |
| Two-Tower | $0.892_{\pm.035}$ | $0.945_{\pm.056}$ | $0.918_{\pm.059}$ | $0.918_{\pm.038}$ | $0.961_{\pm.025}$ |
| + Node intervention | $\mathbf{1.000}_{\pm.000}$ | $\mathbf{1.000}_{\pm.000}$ | $\mathbf{1.000}_{\pm.000}$ | $\mathbf{1.000}_{\pm.000}$ | $\mathbf{1.000}_{\pm.000}$ |
| + Node merging | $0.987_{\pm.000}$ | $\mathbf{1.000}_{\pm.000}$ | $\mathbf{1.000}_{\pm.000}$ | $\mathbf{1.000}_{\pm.000}$ | $\mathbf{1.000}_{\pm.000}$ |

## 5 EXPERIMENTS

### 5.1 FULLY SYNTHETIC DATASETS

**Dataset**    To verify the correctness of Theorem 1, and the effectiveness of proposed methods, we first conducted experiments on a fully synthetic dataset, which allowed for precise control of the connectivity of IGs. We generated four datasets with different numbers $K$ of connected components within each IG ($K = 1, 2, 3, 4$), as illustrated in Appendix E.1. The bias factors only consist with positions (*i.e.*, position-based model or PBM). We defer the click simulation setup to Appendix E.2.

**Baselines**    We tested the following baselines for comparison with our proposed methods: *No debias*, which utilizes click data to train the ranking model without the observation model, and three widely-used ULTR models based on examination hypothesis, *DLA* (Ai et al., 2018a; Vardasbi et al., 2020a; Chen et al., 2021), *Regression-EM* (Wang et al., 2018; Sarvi et al., 2023) and *Two-Tower* (Guo et al., 2019; Chen et al., 2020; 2022a; 2023). Implementation details for these baselines can be found in Appendix E.3. Note that our proposed *node intervention* and *node merging* are model-agnostic and independent of the specific training implementation, which are applied on the datasets before training.

**Evaluation metrics**    To evaluate the performance of the methods, we computed the mean correlation coefficient (MCC) between the original relevance probability $r(\cdot)$ and the predicted relevance probability $r'(\cdot)$, defined as $\sum_{i=1}^{|\mathcal{D}|}\left(r(\boldsymbol{x}_i) - \overline{r(\boldsymbol{x})}\right)\left(r'(\boldsymbol{x}_i) - \overline{r'(\boldsymbol{x})}\right) / \sqrt{\sum_{i=1}^{|\mathcal{D}|}\left(r(\boldsymbol{x}_i) - \overline{r(\boldsymbol{x})}\right)^2}\sqrt{\sum_{i=1}^{|\mathcal{D}|}\left(r'(\boldsymbol{x}_i) - \overline{r'(\boldsymbol{x})}\right)^2}$. A high MCC means that we successfully identified the true relevance model and recovered the true relevance up to a scaling transformation. We also computed nDCG@$k$ ($k = 1, 3, 5, 10$) which are standard measures prevalently used in LTR.

**Analysis: How does the connectivity of IGs impact the ranking performance?**    Figure 2(a) shows the effects of varying numbers of connected components $K$ within IGs on ranking performance (using DLA), with different numbers of clicks. Here, $K = 1$ indicates a connected IG. We can observe that the ranking model is capable of achieving perfect ranking accuracy only if the IG is connected. Otherwise, the performance exhibits significant instability and poor quality, regardless of the number of clicks collected. Besides, larger $K$s (*e.g.*, when $K > 1$) do not give rise to significant differences. These observations serve to validate the correctness of Theorem 1.

**Analysis: Can node intervention and node merging handle unidentifiable dataset?**    We tested node intervention and node merging in the $K = 2$ scenario. Table 1 summarizes the performance. One can see that our proposed two methods achieve consistently perfect ranking accuracy, which significantly outperforms the two baselines. This demonstrates the effectiveness of our methods when the dataset lacks a connected IG. Additionally, node intervention can fully recover the relevance probability with an MCC of 1.0, while node merging cannot. It shows that node merging involves merging two nodes with different observation probabilities, leading to approximation errors. On the other hand, node intervention is lossless but requires an additional operation to augment the dataset.

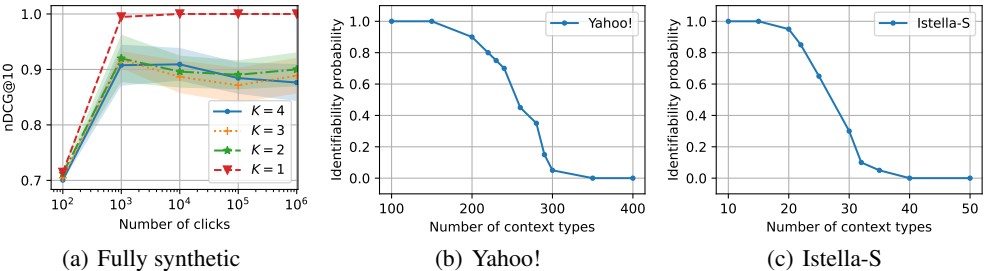

(a) Fully synthetic      (b) Yahoo!      (c) Istella-S

Figure 2: (a) Performance across different number of clicks, on the fully synthetic dataset with different number of connected components $K$. The variance is displayed with the shadow areas. (b) The influence of the number of context types on the identifiability probabilities on the two datasets.

**Ablation studies for node intervention and node merging**   As ablation studies, we utilized different selection strategies for node intervention to verify Proposition 1, and different merging strategies for node merging to verify Proposition 2. Details can be found in Appendix E.4.

## 5.2 LARGE-SCALE STUDY

**Dataset**   We also performed another empirical study on the large-scale semi-synthetic setup that is prevalent in unbiased learning to rank (Joachims et al., 2017; Ai et al., 2021; Chen et al., 2022a) on two widely used benchmark datasets: Yahoo! LETOR (Chapelle & Chang, 2011) and Istella-S (Lucchese et al., 2016). We provide further details for them in Appendix E.1. In both datasets, only the top 10 documents were considered to be displayed. In addition to positions, we also incorporated context types as another bias factor that is prevalent in recent research (*i.e.*, contextual position-based model or CPBM) (Fang et al., 2019; Chen et al., 2021). Random context type was assigned to each query-document pair. Furthermore, we conducted identifiability testing on the TianGong-ST (Chen et al., 2019), a large-scale real-world dataset with an abundance of genuine bias factors.

**Analysis: Does unidentifiability issue exist in real-world?**   We first applied the identifiability check algorithm on TianGong-ST, and found that when accounting for **positions** and all provided **vertical types** as bias factors (a total of 20,704), the IG of this dataset is *disconnected*: there are 2,900 connected components within it. This observation suggests that the unidentifiability phenomenon could occur in real-world, even on a large-scale dataset. We further excluded certain bias factors from that dataset and assessed its identifiability, which are elaborated in Appendix E.5.

**Analysis: How do the number of bias factors and dataset scale affect the identifiability?**   We tuned the number of context types within Yahoo! and Istella-S and computed the frequency with which the IG was connected to determine the identifiability probability. Each experiment was repeated 20 times. From Figure 2(b) and 2(c), it can be observed that if positions are the only bias factors, both datasets are identifiable for the ranking model. However, upon the consideration of context types as bias factors, the identifiability probability drastically decreases as the number of context types increases. Merely 20 (on Istella-S) or 200 (on Yahoo!) are sufficient to render the IG disconnected. When the number is further increased, it becomes exceedingly challenging to obtain an identifiable ranking model, which verifies Corollary 1 and indicates that if we account for too many bias factors, the IGs in real-world scenarios are likely to be disconnected. We also conducted an experiment to investigate the impact of dataset scale on identifiability, elaborated in Appendix E.6.

**Analysis: How does node merging perform under contextual bias?**   We simulated 5,000 context types on Yahoo! and Istella-S to evaluate the efficacy of node merging. In this case, node intervention is not appropriate since the context type cannot be swapped. Table 2 shows the results in both the training and test sets. We can observe that node merging successfully handles the unidentifiable case and recovers the relevance score, outperforming the baselines in terms of both MCC and nDCG on the training set significantly. On the test set, node merging outperforms the baselines on Yahoo!, and achieves comparable ranking performance with DLA on Istella-S. One possible explanation is that

Table 2: Performance (with the standard deviations) comparison on two datasets under CPBM bias.

| | | MCC | training | | test | |
|---|---|---|---|---|---|---|
| | | | nDCG@5 | nDCG@10 | nDCG@5 | nDCG@10 |
| YAHOO! | No Debias | $0.622_{\pm.184}$ | $0.807_{\pm.064}$ | $0.889_{\pm.045}$ | $0.693_{\pm.001}$ | $0.741_{\pm.001}$ |
| | DLA | $0.619_{\pm.003}$ | $0.819_{\pm.001}$ | $0.892_{\pm.001}$ | $0.688_{\pm.001}$ | $0.737_{\pm.001}$ |
| | + Node merging | $\mathbf{0.744_{\pm.002}}$ | $\mathbf{0.863_{\pm.001}}$ | $\mathbf{0.921_{\pm.001}}$ | $\mathbf{0.699_{\pm.001}}$ | $\mathbf{0.746_{\pm.001}}$ |
| ISTELLA-S | No Debias | $0.678_{\pm.135}$ | $0.818_{\pm.099}$ | $0.892_{\pm.069}$ | $0.634_{\pm.001}$ | $0.682_{\pm.001}$ |
| | DLA | $0.748_{\pm.001}$ | $0.874_{\pm.001}$ | $0.931_{\pm.000}$ | $\mathbf{0.638_{\pm.002}}$ | $\mathbf{0.686_{\pm.002}}$ |
| | + Node merging | $\mathbf{0.782_{\pm.001}}$ | $\mathbf{0.900_{\pm.001}}$ | $\mathbf{0.947_{\pm.000}}$ | $\mathbf{0.638_{\pm.001}}$ | $\mathbf{0.686_{\pm.001}}$ |

node merging recovers more accurate relevance scores in the training set, but only a part of them benefit the test set. This phenomenon might be mitigated by expanding training set.

## 6 RELATED WORK

**Unbiased learning to rank (ULTR)**    Debiasing click data using ULTR is a crucial research direction within the field of information retrieval. We review this line of related work in Appendix A.

**Relevance Recovery in ULTR**    The identifiability condition (Theorem 1) establishes connections and generalizations to recent ULTR research. Agarwal et al. (2019c) constructed intervention sets to uncover documents that are put at two different positions to estimate observation probabilities, which, however, did not further explore the recoverable conditions. Oosterhuis (2022) also showed that a perfect click model can provide incorrect relevance estimates and the estimation consistency depends on the data collection procedure. We take a further step by delving into the root cause and digesting the concrete condition based on the data. Zhang et al. (2023) found that some features (*e.g.*, with high relevance) are more likely to be assigned with specific bias factors (*e.g.*, top positions). This phenomenon results in a decline in performance, named as confounding bias. This bias is related to the identifiability issue since when a sever confounding bias is present, the IG is more likely to be disconnected due to insufficient data coverage.

**Identifiability in machine learning**    Identifiability is a fundamental concept in various machine learning fields, such as independent component analysis (Hyvarinen & Morioka, 2017; Hyvarinen et al., 2019), latent variable models (Allman et al., 2009; Guillaume et al., 2019; Khemakhem et al., 2020), missing not at random data (Ma & Zhang, 2021; Miao et al., 2016) and causal discovery (Addanki et al., 2021; Peters et al., 2011; Spirtes & Zhang, 2016). It defines a model's capacity to recover some unique latent structure, variables, or causal relationships from the data. In this work, we embrace the commonly used notion of identifiability and apply its definition to the ULTR domain.

## 7 CONCLUSIONS

In this paper, we take the first step to exploring if and when the relevance can be recovered from click data. We first define the identifiability of a ranking model, which refers to the ability to recover relevance probabilities up to a scaling transformation. Our research reveals that (1) the ranking model is not always identifiable, which depends on the underlying structure of the dataset (*i.e.*, an identifiability graph should be connected); (2) identifiability depends on the data size and the number of bias factors, and unidentifiability issue is possible on large-scale real-world datasets; (3) two methods, node intervention and node merging, can be utilized to address the unidentifiability issues. Our proposed framework are theoretically and empirically verified.

**Limitations**    Our proposed framework is based on the examination hypothesis (Eq.(1)). While this is the most commonly employed generation process for clicks, there exist other assumptions on modeling user's click behavior, such as trust bias (Agarwal et al., 2019b; Vardasbi et al., 2020b; Oosterhuis, 2023) or vectorization-based examination hypothesis (Chen et al., 2022b; Yan et al., 2022). We leave how to develop an identifiability theory in these cases as future work.

REPRODUCIBILITY STATEMENT

The datasets can be downloaded from `https://webscope.sandbox.yahoo.com/` (Yahoo!), `http://quickrank.isti.cnr.it/istella-dataset/` (Istella-S) and `http://www.thuir.cn/tiangong-st/` (TianGong-ST). Codes for proposed algorithms are in supplementary materials. Proofs for the theorems are in Appendix B. Implementation details, experiment settings and additional experimental results are in Appendix E.

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

APPENDIX

# A  EXTENDED RELATED WORK

Unbiased learning to rank (ULTR) tries to directly learn unbiased ranking models from biased clicks. The core of ULTR lies in the estimation of observation probabilities, which is typically achieved through intervention (Wang et al., 2016; Joachims et al., 2017). These methods are related to the node intervention we proposed in § 4.1, but they are prone to useless swapping operations and negatively impact user's experience. To avoid intervention, Agarwal et al. (2019c) and Fang et al. (2019) proposed intervention harvest methods that exploit click logs with multiple ranking models. These methods are related to Theorem 1, but they did not delve into the identifiability conditions. Our proposed theory bridges the gap between these two groups of work by determining when intervention is necessary or not. Recently, some researchers proposed to jointly estimate relevance and bias, including IPS-based methods (Wang et al., 2018; Ai et al., 2018a; Hu et al., 2019; Jin et al., 2020) and two-tower based models (Zhao et al., 2019; Guo et al., 2019; Haldar et al., 2020; Yan et al., 2022). These models are based on the examination hypothesis and are optimized to maximize user click likelihood, therefore our proposed framework can be applied to these models as well.

On the other side, researchers developed models to extend the scope of bias factors that affect observation probabilities, which contain position (Wang et al., 2018; Ai et al., 2018a; Hu et al., 2019; Ai et al., 2021), contextual information (Fang et al., 2019; Tian et al., 2020), clicks in the same query list (Vardasbi et al., 2020a; Chen et al., 2021), presentation style (Zheng et al., 2019; Liu et al., 2015; Chen et al., 2023), search intent (Sun et al., 2020), result domain (Ieong et al., 2012), ranking features (Chen et al., 2022a) and outliers (Sarvi et al., 2023). While incorporating additional bias factors is beneficial for improving the estimation of accurate observation probabilities (Chen et al., 2023), as we mention in Theorem 1 and § 5.2, an excessive number of bias factors may pose a risk of unidentifiability.

# B  PROOFS FOR THE THEORETICAL RESULTS

## B.1  PROOF FOR THEOREM 1

**Step 1.** We first prove the "if" part: assume that

$$r(\boldsymbol{x}) \cdot o(\boldsymbol{t}) = r'(\boldsymbol{x}) \cdot o'(\boldsymbol{t}) \quad \forall (\boldsymbol{x}, \boldsymbol{t}) \in \mathcal{D}, \tag{B.1}$$

and $G$ is connected, our goal is to prove that $r(\boldsymbol{x})/r'(\boldsymbol{x}) = $ constant. Note that we only consider the nontrivial case $r(\boldsymbol{x}) \neq 0$ and $r'(\boldsymbol{x}) \neq 0$. Otherwise, $C$ can be any positive number.

For any two bias factors $\boldsymbol{s} \in \mathcal{T}$ and $\boldsymbol{t} \in \mathcal{T}$, since $G$ is connected, there exists a path $v_{\boldsymbol{a}_1} \to v_{\boldsymbol{a}_2} \to \cdots \to v_{\boldsymbol{a}_n}$ in $G$ where $v_{\boldsymbol{a}_1}, \cdots, v_{\boldsymbol{a}_n}$ are the nodes in the identifiability graph representing different bias factors, and $\boldsymbol{a}_1 = \boldsymbol{s}, \boldsymbol{a}_n = \boldsymbol{t}$. Consider a middle edge $v_{\boldsymbol{a}_m} \to v_{\boldsymbol{a}_{m+1}} (1 \leq m \leq n-1)$, according to the definition of the edge,

$$\exists \boldsymbol{x} \in \mathcal{X}, \text{ s.t. } (\boldsymbol{x}, \boldsymbol{a}_m) \in \mathcal{D} \wedge (\boldsymbol{x}, \boldsymbol{a}_{m+1}) \in \mathcal{D}. \tag{B.2}$$

According to Eq.(B.1) and Eq.(B.2), we have $r(\boldsymbol{x}) \cdot o(\boldsymbol{a}_m) = r'(\boldsymbol{x}) \cdot o'(\boldsymbol{a}_m)$ and $r(\boldsymbol{x}) \cdot o(\boldsymbol{a}_{m+1}) = r'(\boldsymbol{x}) \cdot o'(\boldsymbol{a}_{m+1})$, and therefore

$$\frac{o'(\boldsymbol{a}_m)}{o(\boldsymbol{a}_m)} = \frac{r(\boldsymbol{x})}{r'(\boldsymbol{x})} = \frac{o'(\boldsymbol{a}_{m+1})}{o(\boldsymbol{a}_{m+1})}. \tag{B.3}$$

Let $f(\boldsymbol{x}) = r(\boldsymbol{x})/r'(\boldsymbol{x})$ and $g(\boldsymbol{t}) = o'(\boldsymbol{t})/o(\boldsymbol{t})$. Applying Eq.(B.3) to the path $v_{\boldsymbol{a}_1} \to v_{\boldsymbol{a}_2} \to \cdots \to v_{\boldsymbol{a}_n}$, we obtain $g(\boldsymbol{s}) = g(\boldsymbol{t})$. Given that $\boldsymbol{s}$ and $\boldsymbol{t}$ are selected arbitrarily, we have $g(\boldsymbol{t}) = $ constant for all bias factors $\boldsymbol{t}$. Since $f(\boldsymbol{x}) = g(\boldsymbol{t})$ holds for all $(\boldsymbol{x}, \boldsymbol{t}) \in \mathcal{D}$ according to Eq.(B.1), $f(\boldsymbol{x})$ is also constant.

**Step 2.** We then prove the "only if" part: assume that the relevance is identifiable, prove that $G$ is connected. We prove this by contradiction: Given a disconnected IG $G$, our goal is to prove that the ranking model is unidentifiable.

Since $G = (V, E)$ is disconnected, we suppose $G$ can be divided into two disjoint graphs $G_1 = (V_1, E_1)$ and $G_2 = (V_2, E_2)$. Based on $G_1$ and $G_2$, we can divide the dataset $\mathcal{D}$ into two disjoint sets $D_1$ and $D_2$: $\mathcal{D}_1 = \{(\boldsymbol{x}, \boldsymbol{t}) \mid v_{\boldsymbol{t}} \in V_1\}$ and $\mathcal{D}_2 = \{(\boldsymbol{x}, \boldsymbol{t}) \mid v_{\boldsymbol{t}} \in V_2\}$. Let $\mathcal{X}_1 = \{\boldsymbol{x} \mid (\boldsymbol{x}, \boldsymbol{t}) \in \mathcal{D}_1\}$ denote features in $\mathcal{D}_1$, and $\mathcal{X}_2 = \{\boldsymbol{x} \mid (\boldsymbol{x}, \boldsymbol{t}) \in \mathcal{D}_2\}$ denote features in $\mathcal{D}_2$. Note that $\mathcal{X}_1$ and $\mathcal{X}_2$ are disjoint, *i.e.*, $\mathcal{X}_1 \cap \mathcal{X}_2 = \varnothing$, otherwise according to the definition of the edge set, there exists an edge between $V_1$ and $V_2$ which connects $G_1$ and $G_2$.

Next, given any relevance function $r$ and observation function $o$, we define $r'$ and $o'$ as follows.

$$r'(\boldsymbol{x}) = \begin{cases} \alpha r(\boldsymbol{x}) & \text{if } \boldsymbol{x} \in \mathcal{X}_1, \\ \beta r(\boldsymbol{x}) & \text{if } \boldsymbol{x} \in \mathcal{X}_2, \end{cases}$$

$$o'(\boldsymbol{t}) = \begin{cases} o(\boldsymbol{t})/\alpha & \text{if } v_{\boldsymbol{t}} \in V_1, \\ o(\boldsymbol{t})/\beta & \text{if } v_{\boldsymbol{t}} \in V_2, \end{cases}$$

where $\alpha \neq \beta$ are two positive numbers. Note that if $(\boldsymbol{x}, \boldsymbol{t}) \in \mathcal{D}_1$, then $\boldsymbol{x} \in \mathcal{X}_1$ and $v_{\boldsymbol{t}} \in V_1$, therefore $r'(\boldsymbol{x})o'(\boldsymbol{t}) = \alpha r(\boldsymbol{x}) \cdot o(\boldsymbol{t})/\alpha = r(\boldsymbol{x})o(\boldsymbol{t})$. If $(\boldsymbol{x}, \boldsymbol{t}) \in \mathcal{D}_2$, then $\boldsymbol{x} \in \mathcal{X}_2$ and $v_{\boldsymbol{t}} \in V_2$, therefore $r'(\boldsymbol{x})o'(\boldsymbol{t}) = \beta r(\boldsymbol{x}) \cdot o(\boldsymbol{t})/\beta = r(\boldsymbol{x})o(\boldsymbol{t})$. Based on it, Eq.(B.1) holds for all $(\boldsymbol{x}, \boldsymbol{t}) \in \mathcal{D}$. However, it is obvious that $C$ isn't constant in $r(\boldsymbol{x}) = Cr'(\boldsymbol{x})$, since $C = \alpha$ when $\boldsymbol{x} \in \mathcal{X}_1$ and $C = \beta$ otherwise. It indicates that the relevance model isn't identifiable.

## B.2 PROOF FOR COROLLARY 1

We begin by estimating the disconnected probability between two nodes in the identifiability graph, as the following lemma.

**Lemma B.1.** *In an identifiability graph, the probability of two nodes $v_{\boldsymbol{s}}$ and $v_{\boldsymbol{t}}$ are disconnected can be estimated as:*

$$P(\text{disconnected} \mid \mathcal{D}, \boldsymbol{s}, \boldsymbol{t}) \sim \exp\left(-\frac{|\mathcal{D}|}{|\mathcal{T}|}\right) \left[2 - \exp\left(-\frac{|\mathcal{D}|}{|\mathcal{X}||\mathcal{T}|}\right)\right]^{|\mathcal{X}|},$$

*when $|\mathcal{X}||\mathcal{T}| \to \infty$.*

*Proof.* Let $P(\boldsymbol{x}, \boldsymbol{s})$ and $P(\boldsymbol{x}, \boldsymbol{t})$ denote the probabilities of selecting $(\boldsymbol{x}, \boldsymbol{s})$ and $(\boldsymbol{x}, \boldsymbol{t})$ respectively. We have:

$$
\begin{aligned}
P(\text{disconnected} \mid \mathcal{D}, \boldsymbol{s}, \boldsymbol{t}) &= P\left(\bigcap_{\boldsymbol{x} \in \mathcal{X}} (\boldsymbol{x}, \boldsymbol{s}) \notin \mathcal{D} \vee (\boldsymbol{x}, \boldsymbol{t}) \notin \mathcal{D}\right) \\
&= \prod_{\boldsymbol{x} \in \mathcal{X}} P\left((\boldsymbol{x}, \boldsymbol{s}) \notin \mathcal{D} \vee (\boldsymbol{x}, \boldsymbol{t}) \notin \mathcal{D}\right) \\
&= \prod_{\boldsymbol{x} \in \mathcal{X}} 1 - P\left((\boldsymbol{x}, \boldsymbol{s}) \in \mathcal{D}\right) \cdot P\left((\boldsymbol{x}, \boldsymbol{t}) \in \mathcal{D}\right) \\
&= \prod_{\boldsymbol{x} \in \mathcal{X}} 1 - (1 - P\left((\boldsymbol{x}, \boldsymbol{s}) \notin \mathcal{D}\right)) \cdot (1 - P\left((\boldsymbol{x}, \boldsymbol{t}) \notin \mathcal{D}\right)).
\end{aligned}
$$

Note that $P\left((\boldsymbol{x}, \boldsymbol{t}) \notin \mathcal{D}\right)$ is the probability that $(\boldsymbol{x}, \boldsymbol{t})$ is not sampled for $|\mathcal{D}|$ times, we have $P\left((\boldsymbol{x}, \boldsymbol{s}) \notin \mathcal{D}\right) = [1 - P(\boldsymbol{x}, \boldsymbol{s})]^{|\mathcal{D}|}$ and $P\left((\boldsymbol{x}, \boldsymbol{t}) \notin \mathcal{D}\right) = [1 - P(\boldsymbol{x}, \boldsymbol{t})]^{|\mathcal{D}|}$, therefore,

$$P(\text{disconnected} \mid \mathcal{D}, \boldsymbol{s}, \boldsymbol{t}) = \prod_{\boldsymbol{x} \in \mathcal{X}} 1 - \left(1 - [1 - P(\boldsymbol{x}, \boldsymbol{s})]^{|\mathcal{D}|}\right) \left(1 - [1 - P(\boldsymbol{x}, \boldsymbol{t})]^{|\mathcal{D}|}\right).$$

Using the condition that features and bias factors are sampled independently and uniformly, we have $P(\boldsymbol{x}, \boldsymbol{s}) = P(\boldsymbol{x}, \boldsymbol{t}) = 1/|\mathcal{X}||\mathcal{T}|$. Therefore,

$$
\begin{aligned}
P(\text{disconnected} \mid \mathcal{D}, \boldsymbol{s}, \boldsymbol{t}) &= \left\{ 1 - \left[ 1 - \left( 1 - \frac{1}{|\mathcal{X}||\mathcal{T}|} \right)^{|\mathcal{D}|} \right]^2 \right\}^{|\mathcal{X}|} \\
&= \left\{ 1 - \left[ 1 - \left( 1 - \frac{1}{|\mathcal{X}||\mathcal{T}|} \right)^{-|\mathcal{X}||\mathcal{T}| \cdot \frac{-|\mathcal{D}|}{|\mathcal{X}||\mathcal{T}|}} \right]^2 \right\}^{|\mathcal{X}|} \\
&\sim \left\{ 1 - \left[ 1 - \exp\left( -\frac{|\mathcal{D}|}{|\mathcal{X}||\mathcal{T}|} \right) \right]^2 \right\}^{|\mathcal{X}|} \\
&= \left[ 2\exp\left( -\frac{|\mathcal{D}|}{|\mathcal{X}||\mathcal{T}|} \right) - \exp\left( -\frac{2|\mathcal{D}|}{|\mathcal{X}||\mathcal{T}|} \right) \right]^{|\mathcal{X}|} \\
&= \exp\left( -\frac{|\mathcal{D}|}{|\mathcal{T}|} \right) \left[ 2 - \exp\left( -\frac{|\mathcal{D}|}{|\mathcal{X}||\mathcal{T}|} \right) \right]^{|\mathcal{X}|},
\end{aligned}
$$

where the third line uses $(1 + 1/n)^n \to e$ when $n \to \infty$.

$\square$

We next provide a lemma to estimate the probability that a random graph is connected.

**Lemma B.2.** *(Gilbert, 1959) Suppose a graph $G$ is constructed from a set of $N$ nodes in which each one of the $N(N-1)/2$ possible links is present with probability $p$ independently. The probability that $G$ is connected can be estimated as:*

$$
P(\text{connected} \mid G) \sim 1 - N(1-p)^{N-1}.
$$

Applying Theorem 1, Lemma B.1 and Lemma B.2, we obtain:

$$
\begin{aligned}
P(\text{identifiability} \mid \mathcal{D}) &\sim 1 - |\mathcal{T}| \left\{ \exp\left( -\frac{|\mathcal{D}|}{|\mathcal{T}|} \right) \left[ 2 - \exp\left( -\frac{|\mathcal{D}|}{|\mathcal{X}||\mathcal{T}|} \right) \right]^{|\mathcal{X}|} \right\}^{|\mathcal{T}|-1} \\
&= 1 - |\mathcal{T}| \exp\left[ -|\mathcal{D}| \left( 1 - \frac{1}{|\mathcal{T}|} \right) \right] \left[ 2 - \exp\left( -\frac{|\mathcal{D}|}{|\mathcal{X}||\mathcal{T}|} \right) \right]^{|\mathcal{X}|(|\mathcal{T}|-1)} \\
&\sim 1 - |\mathcal{T}| \exp\left( -|\mathcal{D}| \right) \left[ 2 - \exp\left( -\frac{|\mathcal{D}|}{|\mathcal{X}||\mathcal{T}|} \right) \right]^{|\mathcal{X}||\mathcal{T}|} \\
&= 1 - |\mathcal{T}| \exp\left( -|\mathcal{D}| + |\mathcal{X}||\mathcal{T}| \log\left[ 2 - \exp\left( -\frac{|\mathcal{D}|}{|\mathcal{X}||\mathcal{T}|} \right) \right] \right)
\end{aligned}
$$

where the third line uses $1/|\mathcal{T}| \to 0$ and $|\mathcal{T}| - 1 \to |\mathcal{T}|$ when $|\mathcal{T}|$ is large enough.

### B.3 PROOF FOR PROPOSITION 1

Note that $Nr'(\boldsymbol{x})o'(\boldsymbol{t})$ follows a binomial distribution, *i.e.*,

$$
Nr'(\boldsymbol{x})o'(\boldsymbol{t}) \sim B(N, r(\boldsymbol{x})o(\boldsymbol{t})),
$$

which implies:

$$
\mathbb{E}[Nr'(\boldsymbol{x})o'(\boldsymbol{t})] = Nr(\boldsymbol{x})o(\boldsymbol{t}), \quad \mathbb{V}[Nr'(\boldsymbol{x})o'(\boldsymbol{t})] = Nr(\boldsymbol{x})o(\boldsymbol{t})[1 - r(\boldsymbol{x})o(\boldsymbol{t})].
$$

Denote $g(\boldsymbol{t}) = o'(\boldsymbol{t})/o(\boldsymbol{t})$, then we have:

$$
\begin{aligned}
\mathbb{E}[g(\boldsymbol{t}) \mid r'(\boldsymbol{x})] &= \frac{Nr(\boldsymbol{x})o(\boldsymbol{t})}{Nr'(\boldsymbol{x})o(\boldsymbol{t})} = \frac{r(\boldsymbol{x})}{r'(\boldsymbol{x})}, \\
\mathbb{V}[g(\boldsymbol{t}) \mid r'(\boldsymbol{x})] &= \frac{Nr(\boldsymbol{x})o(\boldsymbol{t})[1 - r(\boldsymbol{x})o(\boldsymbol{t})]}{[Nr'(\boldsymbol{x})o(\boldsymbol{t})]^2} = \frac{r(\boldsymbol{x})(1 - r(\boldsymbol{x})o(\boldsymbol{t}))}{Nr'(\boldsymbol{x})^2o(\boldsymbol{t})}.
\end{aligned}
$$

Since $c(\boldsymbol{x}, \boldsymbol{t})$ are sampled i.i.d., $g(\boldsymbol{t})$ is independent of $g(\boldsymbol{t}')$ conditioned on $r'(\boldsymbol{x})$. Therefore,

$$\mathbb{E}[g(\boldsymbol{t}_1) - g(\boldsymbol{t}_2) \mid r'(\boldsymbol{x})] = \frac{r(\boldsymbol{x})}{r'(\boldsymbol{x})} - \frac{r(\boldsymbol{x})}{r'(\boldsymbol{x})} = 0,$$

$$\mathbb{V}[g(\boldsymbol{t}_1) - g(\boldsymbol{t}_2) \mid r'(\boldsymbol{x})] = \mathbb{V}[g(\boldsymbol{t}_1) \mid r'(\boldsymbol{x})] + \mathbb{V}[g(\boldsymbol{t}_2) \mid r'(\boldsymbol{x})]$$

$$= \frac{r(\boldsymbol{x})(1 - r(\boldsymbol{x})o(\boldsymbol{t}_1))}{Nr'(\boldsymbol{x})^2 o(\boldsymbol{t}_1)} + \frac{r(\boldsymbol{x})(1 - r(\boldsymbol{x})o(\boldsymbol{t}_2))}{Nr'(\boldsymbol{x})^2 o(\boldsymbol{t}_2)}$$

$$= \frac{r(\boldsymbol{x})^2}{Nr'(\boldsymbol{x})^2} \left[ \frac{1 - r(\boldsymbol{x})o(\boldsymbol{t}_1)}{r(\boldsymbol{x})o(\boldsymbol{t}_1)} + \frac{1 - r(\boldsymbol{x})o(\boldsymbol{t}_2)}{r(\boldsymbol{x})o(\boldsymbol{t}_2)} \right]$$

$$= \frac{1}{NR} \left[ \frac{1}{r(\boldsymbol{x})o(\boldsymbol{t}_1)} + \frac{1}{r(\boldsymbol{x})o(\boldsymbol{t}_2)} - 2 \right].$$

### B.4 Proof for Proposition 2

We first separate the dataset $\mathcal{D}$ into two parts: $\mathcal{D}_1$ (corresponding to $G_1$) and $\mathcal{D}_2$ (corresponding to $G_2$), formally,

$$\mathcal{D}_1 = \{(\boldsymbol{x}, \boldsymbol{t}) \in D \mid \boldsymbol{t} \in V_1\},$$
$$\mathcal{D}_2 = \{(\boldsymbol{x}, \boldsymbol{t}) \in D \mid \boldsymbol{t} \in V_2\}.$$

According to Theorem 1, the relevance model $r(\boldsymbol{x})$ ($\boldsymbol{x} \in \{\boldsymbol{x}_i \mid (\boldsymbol{x}_i, \boldsymbol{t}_i) \in \mathcal{D}_1\}$) is identifiable on the dataset $\mathcal{D}_1$, and the relevance model $r(\boldsymbol{x})$ ($\boldsymbol{x} \in \{\boldsymbol{x}_i \mid (\boldsymbol{x}_i, \boldsymbol{t}_i) \in \mathcal{D}_2\}$) is identifiable on the dataset $\mathcal{D}_2$. That is,

$$\frac{r'(\boldsymbol{x}_a)}{r(\boldsymbol{x}_a)} = \frac{r'(\boldsymbol{x}_b)}{r(\boldsymbol{x}_b)}, \quad \forall \boldsymbol{x}_a, \boldsymbol{x}_b \in \{\boldsymbol{x}_i \mid (\boldsymbol{x}_i, \boldsymbol{t}_i) \in \mathcal{D}_1\},$$

$$\frac{r'(\boldsymbol{x}_c)}{r(\boldsymbol{x}_c)} = \frac{r'(\boldsymbol{x}_d)}{r(\boldsymbol{x}_d)}, \quad \forall \boldsymbol{x}_c, \boldsymbol{x}_d \in \{\boldsymbol{x}_i \mid (\boldsymbol{x}_i, \boldsymbol{t}_i) \in \mathcal{D}_2\}. \tag{B.4}$$

Since we have assumed that $\boldsymbol{x}_1$ and $\boldsymbol{x}_2$ appear in $\mathcal{D}$, we can find $\boldsymbol{t}_1$ and $\boldsymbol{t}_2$ such that $(\boldsymbol{x}_1, \boldsymbol{t}_1) \in \mathcal{D}$ and $(\boldsymbol{x}_2, \boldsymbol{t}_2) \in \mathcal{D}$.

(1) If $\boldsymbol{t}_1 \in V_1 \wedge \boldsymbol{t}_2 \in V_1$, or $\boldsymbol{t}_1 \in V_2 \wedge \boldsymbol{t}_2 \in V_2$, then according to Eq.(B.4),

$$\left| \frac{r'(\boldsymbol{x}_1)}{r(\boldsymbol{x}_1)} - \frac{r'(\boldsymbol{x}_2)}{r(\boldsymbol{x}_2)} \right| = 0. \tag{B.5}$$

(2) Otherwise, without loss of generality we suppose $\boldsymbol{t}_1 \in V_1 \wedge \boldsymbol{t}_2 \in V_2$. For $t'$ and $t''$, we can find $\boldsymbol{x}'$ and $\boldsymbol{x}''$ such that $(\boldsymbol{x}', \boldsymbol{t}') \in \mathcal{D}_1$ and $(\boldsymbol{x}'', \boldsymbol{t}'') \in \mathcal{D}_2$. According to Eq.(B.4),

$$\frac{r'(\boldsymbol{x}_1)}{r(\boldsymbol{x}_1)} = \frac{r'(\boldsymbol{x}')}{r(\boldsymbol{x}')}, \quad \frac{r'(\boldsymbol{x}_2)}{r(\boldsymbol{x}_2)} = \frac{r'(\boldsymbol{x}'')}{r(\boldsymbol{x}'')}.$$

Since

$$\frac{r'(\boldsymbol{x}')}{r(\boldsymbol{x}')} = \frac{o(\boldsymbol{t}')}{o'(\boldsymbol{t}')}, \quad \frac{r'(\boldsymbol{x}'')}{r(\boldsymbol{x}'')} = \frac{o(\boldsymbol{t}'')}{o'(\boldsymbol{t}'')},$$

we have

$$\left| \frac{r'(\boldsymbol{x}_1)}{r(\boldsymbol{x}_1)} - \frac{r'(\boldsymbol{x}_2)}{r(\boldsymbol{x}_2)} \right| = \left| \frac{o(\boldsymbol{t}')}{o'(\boldsymbol{t}')} - \frac{o(\boldsymbol{t}'')}{o'(\boldsymbol{t}'')} \right| = \left| \frac{o(\boldsymbol{t}') - o(\boldsymbol{t}'')}{o'(\boldsymbol{t}')} \right|, \tag{B.6}$$

where we use the fact that $o'(\boldsymbol{t}') = o'(\boldsymbol{t}'')$.

Combining Eq.(B.5) and Eq.(B.6) we obtain the desired result.

## B.5 ERROR BOUND FOR NODE MERGING

Suppose an IG consists of $K$ connected components $\{G_i = (V_i, E_i)\}_{i=1}^{K}$. A node merging algorithm merges $K-1$ pairs of nodes $\mathcal{T} = \{(\boldsymbol{t}_{a_i}, \boldsymbol{t}_{b_i})\}_{i=1}^{K-1}$, forcing $o'(\boldsymbol{t}_{a_i}) = o'(\boldsymbol{t}_{b_i})$. Based on it, we construct the following weighted connected graph $\mathcal{G} = (\mathcal{V}, \mathcal{E})$ built on $K$ components:

$$
\mathcal{V} = \{G_1, G_2, \cdots, G_K\},
$$
$$
\mathcal{E} = \{(G_i, G_j, w) \mid \exists (\boldsymbol{t}_i, \boldsymbol{t}_j) \in \mathcal{T}, \text{s.t.}, \boldsymbol{t}_i \in V_i, \boldsymbol{t}_j \in V_j, w := |o(\boldsymbol{t}_i) - o(\boldsymbol{t}_j)/o'(\boldsymbol{t}_i)|\},
$$

where $(G_i, G_j, w)$ denotes an edge connecting $(G_i, G_j)$ with a weight $w$.

Based on the above notation, we derive the following error bound for node merging.

**Corollary 2** (Error bound of node merging).

$$
r(\boldsymbol{x}) \cdot o(\boldsymbol{t}) = r'(\boldsymbol{x}) \cdot o'(\boldsymbol{t}) \quad \Longrightarrow \quad \left| \frac{r'(\boldsymbol{x}_1)}{r(\boldsymbol{x}_1)} - \frac{r'(\boldsymbol{x}_2)}{r(\boldsymbol{x}_2)} \right| \leq \text{LONGEST-PATH}(\mathcal{G}),
$$

*where* LONGEST-PATH$(\mathcal{G})$ *is the maximum of the sum of the weights for each path in* $\mathcal{G}$.

*Proof.* This is a direct corollary by applying Proposition 2 and the triangle inequality to the edges of each path in $\mathcal{G}$. $\qquad\square$

## C ILLUSTRATIONS OF NODE INTERVENTION AND NODE MERGING

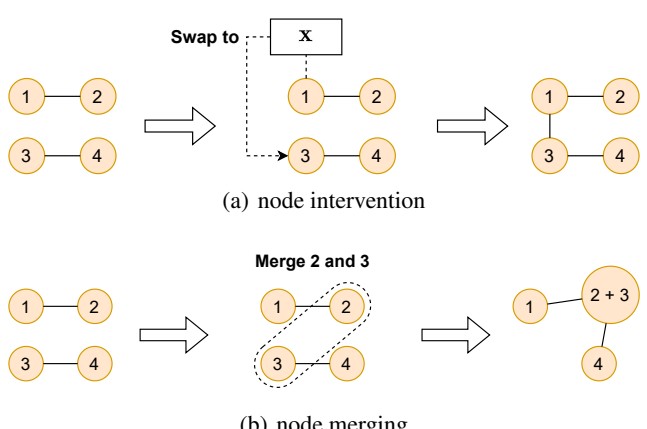

(a) node intervention

(b) node merging

Figure 3: Illustrations for two methods.

Figure 3 illustrates examples for node intervention and node merging. In node intervention, we swap a feature $\mathbf{x}$ related to node 1 (*i.e.*, the 1st bias factor) to node 3 (*i.e.*, the 3rd bias factor) which connects node 1 and node 3. In node merging, we merge node 2 and node 3 into a new node 2 + 3, which indicates that the 2nd bias factor and 3rd bias factor will have a same observation probability.

## D ALGORITHMS

### D.1 IDENTIFIABILITY CHECK

Based on Theorem 1, we illustrate the identifiability check in Algorithm 1. In lines 1-8, we construct a mapping, from feature to the bias factor sets that ever appear together with it. In lines 9-15, we connect bias factor set for each feature to a complete graph, since they are all related to a same feature and thus connect to each other.

---

**Algorithm 1:** Identifiability check

**Input:** Dataset $\mathcal{D} = \{(\boldsymbol{x}_i, \boldsymbol{t}_i)\}_{i=1}^{|\mathcal{D}|}$
**Output:** Whether a relevance model trained on $\mathcal{D}$ is identifiable

1  $S \leftarrow$ Dictionary() ;                    // Initialize $S$ with an empty dictionary
2  **for** $i = 1$ **to** $|\mathcal{D}|$ **do**    // Construct a mapping:  feature $\rightarrow$ bias factors list
3  |   **if** $\boldsymbol{x}_i \notin S$ **then**
4  |   |   $S[\boldsymbol{x}_i] \leftarrow \{\boldsymbol{t}_i\}$;
5  |   **else**
6  |   |   $S[\boldsymbol{x}_i] \leftarrow S[\boldsymbol{x}_i] \cup \{\boldsymbol{t}_i\}$;
7  |   **end**
8  **end**
9  $V \leftarrow \{v_1, v_2, \cdots, v_{|\mathcal{T}|}\}$;
10  $E \leftarrow \varnothing$;
11  **for** $\boldsymbol{x} \in S$ **do**       // Construct identifiability graph (IG) based on $S$
12  |   **for** $\boldsymbol{t}_1, \boldsymbol{t}_2 \in S[\boldsymbol{x}] \times S[\boldsymbol{x}]$ **do**
13  |   |   $E \leftarrow E \cup \{(v_{\boldsymbol{t}_1}, v_{\boldsymbol{t}_2})\}$;
14  |   **end**
15  **end**
16  **if** $G = (V, E)$ *is connected* **then**
17  |   **return** *true*
18  **else**
19  |   **return** *false*
20  **end**

---

**Algorithm 2:** Node intervention

**Input:** Dataset $\mathcal{D} = \{(\boldsymbol{x}_i, \boldsymbol{t}_i)\}_{i=1}^{|\mathcal{D}|}$
**Output:** Intervention set $E'$

1  Construct the IG $G = (V, E)$ on $\mathcal{D}$ using Algorithm 1;
2  $U \leftarrow \{G_1 = (V_1, E_1), \cdots, G_K = (V_K, E_K)\}$ denoting $K$ connected components for $G$;
3  $E' \leftarrow \{\}$ ;                                  // Initialize intervention set
4  $U' \leftarrow \{G_1\}$ ;          // Initialize found nodes for Prim's algorithm
5  **while** $|U'| \neq K$ **do**              // Construct a MST using Prim's algorithm
6  |   $c_{\min} = +\infty$;
7  |   **for** $G_i \in U - U', G_j \in U'$ **do**
8  |   |   Compute $\boldsymbol{x}^{(i,j)}, \boldsymbol{t}^{(i,j)}$ and the intervention cost $c$ based on Eq.(3) - Eq.(6);
9  |   |   **if** $c < c_{\min}$ **then**
10 |   |   |   $c_{\min} \leftarrow c$;
11 |   |   |   $\boldsymbol{x}^*, \boldsymbol{t}^* \leftarrow \boldsymbol{x}^{(i,j)}, \boldsymbol{t}^{(i,j)}$;
12 |   |   |   $G^* \leftarrow G_j$;
13 |   |   **end**
14 |   **end**
15 |   $E' \leftarrow E' \cup \{(\boldsymbol{x}^*, \boldsymbol{t}^*)\}$ ;                  // Add the best intervention pair
16 |   $U' \leftarrow U' \cup \{G^*\}$ ;      // Update found nodes using Prim's algorithm
17 **end**
18 **return** $E'$

### D.2 FULL ALGORITHM FOR NODE INTERVENTION

We illustrate the full algorithm for node intervention (§ 4.1) in Algorithm 2.

Here we use Prim's algorithm to find the MST. In lines 1-2, we construct the IG and find $K$ connected components. In line 3, we initialize the intervention set. In line 4, we initialize the found node set to the first connected components for running Prim's algorithm. In line 7, we traverse the components in the unfound set $U - U'$ (denoted by $G_i$) and in the found set $U'$ (denoted by $G_j$), and compute the intervention cost and the best intervention pair between $G_i$ and $G_j$ in line 8. If the cost is the best, we record the cost, intervention pair and the target component in lines 10-12. Finally, we add the best intervention pair we found in line 15, and update the found set in line 16 for Prim's algorithm.

### D.3 FULL ALGORITHM FOR NODE MERGING

We illustrate the full algorithm for node merging (§ 4.2) in Algorithm 3.

---

**Algorithm 3:** Node merging

**Input:** Dataset $\mathcal{D} = \{(\boldsymbol{x}_i, \boldsymbol{t}_i)\}_{i=1}^{|\mathcal{D}|}$
**Output:** Merging set $E'$

1 Construct the IG $G = (V, E)$ on $\mathcal{D}$ using Algorithm 1;
2 $U \leftarrow \{G_1 = (V_1, E_1), \cdots, G_K = (V_K, E_K)\}$ denoting $K$ connected components for $G$;
3 $E' \leftarrow \{\}$ ;            // Initialize merging set
4 $U' \leftarrow \{G_1\}$ ;      // Initialize found nodes for Prim's algorithm
5 **while** $|U'| \neq K$ **do**       // Construct a MST using Prim's algorithm
6     $c_{\min} = +\infty$;
7     **for** $G_i \in U - U', G_j \in U'$ **do**
8        Compute $\boldsymbol{t}_i^*, \boldsymbol{t}_j^*$ and the merging cost $c$ on Eq.(7) - Eq.(8);
9        **if** $c < c_{\min}$ **then**
10           $c_{\min} \leftarrow c$;
11           $\boldsymbol{t}_A^*, \boldsymbol{t}_B^* \leftarrow \boldsymbol{t}_i^*, \boldsymbol{t}_j^*$;
12           $G^* \leftarrow G_j$;
13        **end**
14     **end**
15     $E' \leftarrow E' \cup \{(\boldsymbol{t}_A^*, \boldsymbol{t}_B^*)\}$ ;          // Add the best merging pair
16     $U' \leftarrow U' \cup \{G^*\}$ ;     // Update found nodes using Prim's algorithm
17 **end**
18 **return** $E'$

---

Similar to node intervention, here we also use Prim's algorithm to find the MST. In lines 1-2, we construct the IG and find $K$ connected components. In line 3, we initialize the merging set. In line 4, we initialize the found node set to the first connected components for running Prim's algorithm. In line 7, we traverse the components in the unfound set $U - U'$ (denoted by $G_i$) and in the found set $U'$ (denoted by $G_j$), and compute the merging cost and the best intervention pair between $G_i$ and $G_j$ in line 8. If the cost is the best, we record the cost, merging pair and the target component in lines 10-12. Finally, we add the best merging pair we found in line 15, and update the found set in line 16 for Prim's algorithm.

Table 3: Dataset statistics

|                    | Yahoo!  | Istella-S |
|--------------------|---------|-----------|
| # Queries          | 28,719  | 32,968    |
| # Documents        | 700,153 | 3,406,167 |
| # Features         | 700     | 220       |
| # Relevance levels | 5       | 5         |

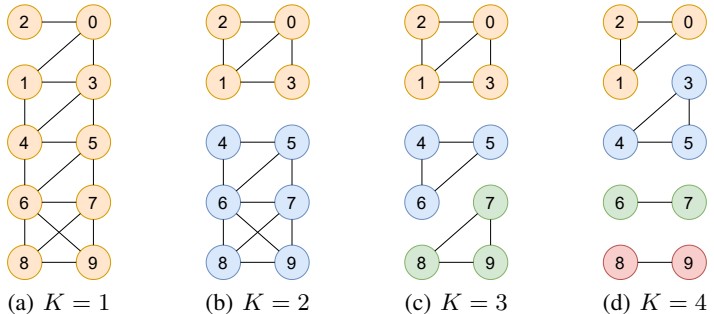

Figure 4: IGs of the simulated datasets. Numbers in the nodes denote the position index (starting from 0 to 9).

Table 4: Identifiability graph statistics

|  | Yahoo! | Istella-S |
|---|---|---|
| # Nodes | 48,894 | 48,919 |
| # Edges | 59,811,898 | 5,972 |
| # Connected components (CC) | 28,958 | 47,976 |
| # Nodes in the Top 1 CC | 15,684 | 101 |
| # Nodes in the Top 2 CC | 98 | 7 |
| # Nodes in the Top 3 CC | 96 | 5 |

# E  EXPERIMENTS

## E.1  DATASETS

In this section, we show details about the datasets we used in this work, including the simulated datasets and semi-synthetic datasets.

**Further details of the simulated datasets**  For fair comparison, all simulated datasets comprised of 10,000 one-hot encoded documents and 1,150 queries with randomly sampled 5-level relevance, and each query contains 10 documents. Figure 4 demonstrates the IGs we used for simulating datasets, where the number of connected components are $K = 1, 2, 3, 4$ respectively.

**Further details of the semi-synthetic datasets**  We followed the given data split of training, validation and testing. To generate initial ranking lists for click simulation, we followed the standard process (Joachims et al., 2017; Ai et al., 2018a; Chen et al., 2021; 2022a) to train a Ranking SVM model (Joachims, 2006) with 1% of the training data with relevance labels, and sort the documents. We used ULTRA framework (Ai et al., 2018b; 2021) to pre-process datasets. Table 3 shows the characteristics of the two datasets we used.

Since the IGs of the two datasets we used are too large to visualize, we show several graph characteristics about them in Table 4 where the number of context types is 5,000.

## E.2  CLICK SIMULATION

**Position-based model**  We sampled clicks according to the examination hypothesis (Eq.(1)) for fully simulated datasets. Following the steps proposed by Chapelle et al. (2009), we set the relevance probability to be:

$$r(\boldsymbol{x}) = \epsilon + (1 - \epsilon)\frac{2^{y_{\boldsymbol{x}}} - 1}{2^{y_{\max}} - 1}, \tag{E.1}$$

where $y_{\boldsymbol{x}} \in [0, y_{\max}]$ is the relevance level of $\boldsymbol{x}$, and $y_{\max} = 4$ in our case. $\epsilon$ is the click noise level and we set $\epsilon = 0.1$ as the default setting. For the observation part, following Ai et al. (2021) we

adopted the position-based examination probability $o(p)$ for each position $p$ by eye-tracking studies (Joachims et al., 2005).

**Contextual position-based model** For simulating contextual bias on the semi-synthetic dataset, following Fang et al. (2019), we assigned each context id $t$ with a vector $\boldsymbol{X}_t \in \mathbb{R}^{10}$ where each element is drawn from $\mathcal{N}(0, 0.35)$. We followed the same formula as position-based model for click simulation, while the observation probability takes the following formula:

$$o(t, p) = o(p)^{\max\{\boldsymbol{w}^\top \boldsymbol{X}_t + 1, 0\}},$$

where $o(p)$ is the position-based examination probability used in the fully synthetic experiment. $\boldsymbol{w}$ is fixed to a 10-dimensional vector uniformly drawn from $[-1, 1]$.

### E.3 TRAINING DETAILS

**Implementation details of baselines** *DLA* (Ai et al., 2018b) uses the following formula to learn the relevance model $r'$ and observation model $o'$ dually:

$$r'_k(\boldsymbol{x}) \leftarrow \arg\min_{r'(\boldsymbol{x})} \sum_{i=1}^{|\mathcal{D}|} \mathbb{1}_{\boldsymbol{x}_i = \boldsymbol{x}} (c_i - o'_{k-1}(\boldsymbol{t}_i) r'(\boldsymbol{x}))^2,$$

$$o'_k(\boldsymbol{t}) \leftarrow \arg\min_{o'(\boldsymbol{t})} \sum_{i=1}^{|\mathcal{D}|} \mathbb{1}_{\boldsymbol{t}_i = \boldsymbol{t}} (c_i - o'(\boldsymbol{t}) r'_{k-1}(\boldsymbol{x}_i))^2,$$

where $\mathcal{D} = \{(\boldsymbol{x}_i, \boldsymbol{t}_i, c_i)\}_{i=1}^{|\mathcal{D}|}$ is the dataset, $r'_k(\boldsymbol{x})$ and $o'_k(\boldsymbol{t})$ are the $k$-th step model output ($1 \leq k \leq T$). The initial values for $o'_0$ and $r'_0$ were randomly initialized from a uniform distribution within the range of $[0, 1]$. After each step, we applied a clipping operation to constrain the outputs within the interval $[0, 1]$.

*Regression-EM* (Wang et al., 2018) uses a iterative process similar to DLA, while the relevance model $r'$ and observation model $o'$ are learned as follows:

$$r'_k(\boldsymbol{x}) \leftarrow \frac{\sum_{i=1}^{|\mathcal{D}|} \mathbb{1}_{\boldsymbol{x}_i = \boldsymbol{x}} \left\{ c_i + (1 - c_i) \frac{[1 - o'_{k-1}(\boldsymbol{t}_i)] r'_{k-1}(\boldsymbol{x}_i)}{1 - o'_{k-1}(\boldsymbol{t}_i) r'_{k-1}(\boldsymbol{x}_i)} \right\}}{\sum_{i=1}^{|\mathcal{D}|} \mathbb{1}_{\boldsymbol{x}_i = \boldsymbol{x}}},$$

$$o'_k(\boldsymbol{t}) \leftarrow \frac{\sum_{i=1}^{|\mathcal{D}|} \mathbb{1}_{\boldsymbol{t}_i = \boldsymbol{t}} \left\{ c_i + (1 - c_i) \frac{[1 - r'_{k-1}(\boldsymbol{x}_i)] o'_{k-1}(\boldsymbol{t}_i)}{1 - o'_{k-1}(\boldsymbol{t}_i) r'_{k-1}(\boldsymbol{x}_i)} \right\}}{\sum_{i=1}^{|\mathcal{D}|} \mathbb{1}_{\boldsymbol{t}_i = \boldsymbol{t}}}.$$

*Two-Tower* (Guo et al., 2019) treats $r'$ and $o'$ as two towers and facilitate the multiplication of the outputs of them close to clicks. They use the binary cross entropy loss to train the models by gradient descent, formulated as:

$$\mathcal{L} = \sum_{i=1}^{|\mathcal{D}|} -c_i \log[r'(\boldsymbol{x}_i) o'(\boldsymbol{t}_i)] - (1 - c_i) \log[1 - r'(\boldsymbol{x}_i) o'(\boldsymbol{t}_i)],$$

where $r'$ and $o'$ are constrained to $[0, 1]$ by applying a sigmoid function.

**Hyper-parameters** We run each experiment for 10 times and reported the average values as well as the standard deviations. On the fully synthetic datasets, we implemented the ranking and observation models as embedding models, and controlled $T = 20,000$ to ensure the convergence. On the semi-synthetic datasets, we also implemented the ranking and observation models as embedding models by assigning a unique identifier based on ranking features to each document, which improves the model's ability to fit clicks during training. The number of epochs was $T = 100$. After training, in order to generalize to the test set, we trained a LightGBM (Ke et al., 2017) as a ranking model with the learned relevance embeddings of each features. The total number of trees was 500, learning rate was 0.1, number of leaves for one tree was 255.

**Implementation details of node merging** For node merging, we used the position number as the bias features on the fully synthetic dataset. On the semi-synthetic dataset, we formed the bias feature $X_{p,t}$ for each bias (consisting of position $p$ and context type $t$) as follows: we multiplied $p$ by 10 and added it to the end of the 10-dimensional context vector $X_t$, to form a 11-dimensional bias feature. This method increases the weight of position, forcing node merging to give priority to merging different context types rather than positions.

**Implementation details of node intervention** For node intervention on the fully synthetic dataset, we trained the ranking model and observation model using node merging, and used their values to implement the cost function (Eq.(3)).

### E.4   ABLATION STUDIES FOR NODE INTERVENTION AND NODE MERGING

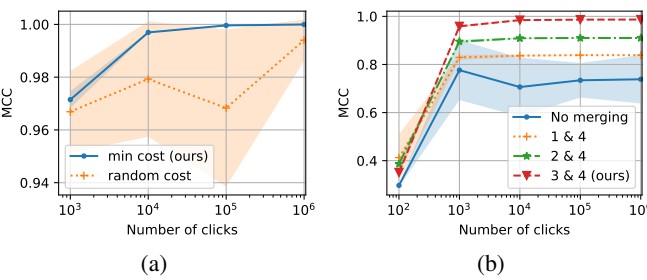

(a)                                        (b)

Figure 5: Performance across different number of clicks. The variance is displayed with the shadow areas. (a) the influence of cost selection strategies in node intervention. (b) the influence of merging strategies in node merging.

**Analysis: The influence of different selection strategies for node intervention.** We conducted an ablation study for the node intervention method in the $K = 2$ scenario. Specifically, we tested a variant named *random cost* by enforcing the cost function (Eq.(3)) follows a uniform distribution on $[0, 1]$, which is independent of $x$ and $t$. This setup leads to an arbitrary selection of the intervention pair (Eq.(4)-Eq.(6)). The initial method was marked as *min cost*. We can observe in Figure 5(a) that, the variance of *random cost* strategy is significantly greater than that of *min cost*. It requires sufficient clicks to obtain a stable performance. This confirms the validity of Theorem 1. Besides, random cost can still achieve a satisfactory level of performance compared to examination hypothesis running on an unidentifiable dataset, demonstrating the importance of a connected IG once again.

**Analysis: The influence of different merging strategies for node merging.** Similarly, we conducted an ablation study for the node merging method in the $K = 2$ scenario. We use three different merging strategies, where $a\&b$ represents merging nodes corresponding to the position $a$ and $b$: (1) 1&4, (2) 2&4, and (3) 3&4. Note that all of the strategies ensure a connected IG, and 3&4 is the proposed node merging strategy. As shown in Figure 5(b), the closer the merging nodes are, the better the performance, which verifies Theorem 2.

### E.5   FURTHER INVESTIGATION ON THE IDENTIFIABILITY OF TIANGONG-ST

On the TianGong-ST dataset, vertical types are represented in the format "$v_1 \# v_2$", *e.g.*, "-1#-1" or "30000701#131". We investigated the consequences of excluding specific bias factors, for example, disregarding either $v_1$ or $v_2$. A summary of the results is presented in Table 5. Our findings reveal that when one of $v_1$ and $v_2$ is kept, the IG loses its connectivity, demonstrating the prevalence of unidentifiability issues in real-world scenarios. However, when only the position is retained, the IG regains connectivity, rendering it identifiable. We observed the reason is that some queries are repeated in the dataset, with variations in the order of related documents. This observation suggests that the search engine may have done some position intervention during deployment which enhances the IG's connectivity. Overall, it reveals again that introducing excessive bias factors leads to a higher probability of unidentifiability.

Table 5: Identifiability of TianGong-ST in different bias factor settings.

| Bias factor | # Connected components | Identifiable? |
|---|---|---|
| $(v_1, v_2, \text{position})$ | 2,900 | $\times$ |
| $(v_1, \text{position})$ | 1,106 | $\times$ |
| $(v_2, \text{position})$ | 87 | $\times$ |
| position only | 1 | $\checkmark$ |

### E.6 THE INFLUENCE OF THE SAMPLING RATIO / NUMBER OF FEATURES ON THE IDENTIFIABILITY PROBABILITIES

We sampled the datasets randomly according to a sampling ratio for 20 times and calculated the frequency that the IG calculated on the sampled datasets is connected, when *positions are the only bias factors*. From Figure 6(a) and 6(b), one can find that although the IGs on both the entire datasets are connected, the subsets of the datasets are not. Specifically, the connectivity of IGs can be guaranteed only when the size of the subset comprises more than approximately 2% (in the case of Yahoo!) and 50% (in the case of Istella-S) of the respective dataset.

We next sampled the features randomly for 20 times, to verify its influence on the probability of identifiability. The number of bias factors is 500 for Yahoo!, and 50 for Istella-S. Figure 6(c) and 6(d) show the results. We can observe that the probability of identifiability decreases as the number of features increases.

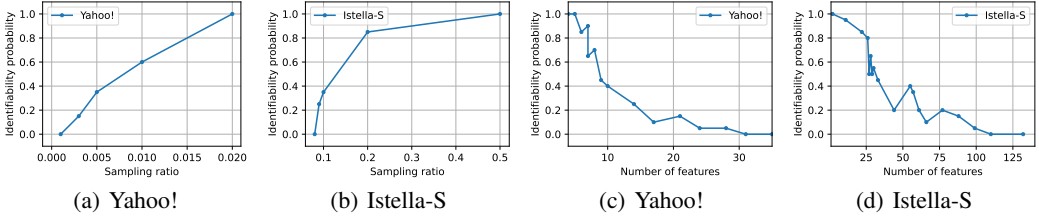

| (a) Yahoo! | (b) Istella-S | (c) Yahoo! | (d) Istella-S |

Figure 6: The influence of (a)(b) the sampling ratio and (c)(d) number of features of datasets on the identifiability probabilities, on Yahoo! and Istella-S, respectively.

