# OpenReview forum: "Identifiability Matters: Revealing the Hidden Recoverable Condition in Unbiased Learning to Rank"
_ICLR.cc/2024/Conference — Submitted to ICLR 2024_

### Official Review · Reviewer_CYAM · 2023-10-30

**Soundness:** 1 poor
**Presentation:** 3 good
**Contribution:** 2 fair
**Rating:** 3
**Confidence:** 4

**Summary:**

The submission studies the identifiability problem of unbiased learning to rank (ULTR) - given a dataset of implicit feedback, whether the true relevance can be identified or not. By treating each bias configuration as a node, and the shared input feature vector for the edges, the relevance model is identifiable if and only if the graph is connected. Then two methods are proposed to try to address the issues (making the graph connected). Experiments are conducted on two synthetic datasets and an offline dataset, where it shows that using the two methods to change data can improve the performance of a naive baseline.

**Strengths:**

It is interesting to study the identifiability problem of unbiased learning to rank from the graph connectivity perspective, though the reviewer is not convinced that this is "the first work" to study the identifiability issue given existing work on coupling / confounding, etc. The major novelty seems to be the graph view.

The proposed two methods are easy to understand. The authors do acknowledge the caveats of the proposed methods (one being not very practical and one may propagate errors).

The paper is overall clearly written.

**Weaknesses:**

Overall the paper is not satisfactory in term of theories or experiments.

Though the theorems look interesting at first glance, the reviewer feels they are generally not very solid or practical after a closer look at them. A major issue is the reviewer feels the submission has self-contradictions in several places.

So there are two scenarios in practice: 1) there are a lot of data, and the bias factor space is small. This is the a common case in practice and people are just fine without worrying about identifiability. The analysis and methods in this paper mostly do not apply since the graph is small and likely connected. 2) There are a lot of bias factors and the graph is more likely to be not connected. The paper mainly argues about this scenario. So far so good and it is ok to focus on 2).

However, by closing look at each theorem, all of them are questionable and some look contradictory to the focus/motivation:

Theorem1: while the trend of using more bias factors is the trend is debatable (especially given existing work showing the coupling / confounding effect), the trend to enrich x is clear. Many real-world applications have personalized feature vectors - then the graph is not likely to be connected even with a small number of bias factors. The paper does not concern this perspective, also, people are fine working with ULTR with such datasets. This questions the value of the proposed framework - one should also note that the condition is only a sufficient condition.

Theorem 2: The assumptions are too strong to make meaningful value from this theorem. The reviewer understands this is to show a simplified “estimate”, but still the value is quite limited for this highly practical field.

Theorem 3: There’s self-contradiction with the motivation of the paper. As discussed, the paper mainly concerns large bias factor space. However, this theorem assumes that each (input feature , bias factor) pair need to sample N observations from a Bernoulli distribution, and the theorem is based on “N is sufficiently large”. How can this be meaningful under the scenario the paper is concerned with?

Theorem 4: The error bound is only shown to merge two subgraphs. Again, the paper is concerned with large bias factor space and the number of subgraphs could be be high - what is the error bound for the entire merge process? Will the errors propagate to meaningless values? Showing error bound only for merging two subgraphs looks quite limited.

There are also several places in the paper that also look contradictory, e.g. when it argues about the benefit of node intervention, “It should be noted that K is typically smaller the the number of positions (assuming that positions are the sole bias factor)” - the reviewer is confused about such claims. If the paper is concerned with such scenario, then there’s probably no need to worry about the identifiability issue.

On the experiments part, the evaluation is weak. The major issue is, the proposed methods are only shown to improve the very basic baseline. The authors argue that the methods are agnostic to the actual algorithm and “aptly represents the current research” - the reviewer strongly disagrees - to show the proposed method is really meaningful,  it needs to show that they can help more sensible baselines. For example, will they help state-of-the-art ULTR methods? If not, why would people care? This is important since the proposed two methods have clear caveats (the node intervention method is not very practical, the node merging method is likely to introduce errors).

**Questions:**

See questions above.

---

> ### Author Response · Authors · 2023-11-13
> **Official Comment by Authors (1/2)**
>
> Thank your for your detailed and valuable feedback. Below are our responses to the weaknesses.
>
> > Theorem 1: Many real-world applications have personalized feature vectors - then the graph is not likely to be connected even with a small number of bias factors ... people are fine working with ULTR with such datasets ... one should also note that the condition is only a sufficient condition.
>
> - We appreciate your insightful suggestion about the feature count. We completely agree that the graph is not likely to be connected when enriching $X$. To demonstrate it, we conducted a thorough experiment for tuning the feature numbers, confirming that increasing X can indeed lead to potential unidentifiability issues. The results are in Appendix E.6 in the revised version. **The current work is okay in public ULTR datasets because the number of features hasn't reached the unidentifiability threshold, as we showed in Section 5.2.**
>
> - We would like to kindly highlight that Theorem 1 is not just a sufficient condition. It's a **necessary and sufficient** condition for identifiability and relevance recovery.
>
> > Theorem 2: The assumptions are too strong to make meaningful value from this theorem. The reviewer understands this is to show a simplified “estimate”, but still the value is quite limited for this highly practical field.
>
> Thanks for pointing it out. Note that this theorem is not critical for our proposed method and the experiments we conducted. Its purpose is to provide an intuition of how "increasing bias factors or decreasing data can potentially lead to unidentifiability" in a simple scenario. We've verified this in a more complex and practical setting in Section 5.2. **To reduce its significance, we've renamed Theorem 2 to Corollary 1.**
>
> > Theorem 3: There’s self-contradiction with the motivation of the paper ... the theorem is based on “N is sufficiently large”.
>
> Thanks for pointing it out. After careful inspection, we've discovered that the assumption "N is sufficiently large" is not necessary for our proposed method, as it's only used for applying the central limit theorem. **We have removed this unnecessary assumption and restructured the theorem to present only the variance term, which is all our method requires.**
>
> > Theorem 4: The error bound is only shown to merge two subgraphs.
>
> Apologies for the confusion about the theorem name "error bound of merging". Its purpose is not to establish the error bound for node merging but to illustrate the error when combining two components, which forms the foundation for the below merging cost (Eq.7). **To address the error bound for node merging, we have created another corollary in Appendix B.5.**
>
> > “It should be noted that K is typically smaller the the number of positions (assuming that positions are the sole bias factor)” ... there’s probably no need to worry about the identifiability issue.
>
> This is because most of current existing ULTR intervention research (Joachims et al., 2017; Radlinski & Joachims, 2006; Carterette & Chandar, 2018; Yue et al., 2010) swap documents between positions. To make fair comparisons, we use positions as an example and show that our method involves far fewer swaps compared to these intervention methods. As you mentioned earlier, increasing X might lead to an unidentifiability issue even if the number of bias factors isn't large. Therefore studying identifiability in such case is still valuable.
>
> > Experiment: will they help state-of-the-art ULTR methods?
>
> We appreciate your feedback about the evaluation. **We've made changes to the paper, and now it includes results for DLA, Regression-EM, and Two-Tower**. Please check our general response for more details.
>
> ---
>
> *To be continued*

---

> ### Author Response · Authors · 2023-11-13
> **Official Comment by Authors and Awaiting for Further Response (2/2)**
>
> (*Continuing from the previous comment*)
>
> ---
>
> Below are the responses for your other reviews:
>
> > The reviewer is not convinced that this is "the first work" to study the identifiability issue given existing work on coupling / confounding, etc
>
> To the best of our knowledge, **traditional approaches doesn't give a necessary and sufficient condition for relevance recovery without specific model constraint**. Previous work on coupling [1] and confounding [2] demands significant changes to the observation model structure, which limits their application in all cases. Our approach from a graph connectivity standpoint offers a principled solution: when the graph is connected, recovery is possible, but when it's not, there's a risk of unrecoverable cases. If you know about any other related work, feel free to share with us.
>
> [1] LBD: Decouple relevance and observation for individual-level unbiased learning to rank.
>
> [2] Towards disentangling relevance and bias in unbiased learning to rank.
>
> > So there are two scenarios in practice: 1) there are a lot of data, and the bias factor space is small. The analysis and methods in this paper mostly do not apply in such case ...
>
> We acknowledge that node intervention/merging are indeed unnecessary in the identifiability issue. However, **we would like to kindly highlight that Theorem 1 (identifiability checking) is also our central contribution and important analysis**, which builds a theoretical base for current ULTR algorithms and is applicable to all cases. As R ggp4 said, this graph view is useful for nicer properties, more intuition-based understandings and more explainability in ULTR. This might provide a new perspective for future ULTR research.
>
> ---
>
> We hope the above responses and revisions could address your concerns. **We genuinely appreciate your insights and are open to hearing which specific parts you still find contradictory.** If you have further questions or any additional concerns, please don't hesitate to ask, and we'll do our best to address them. We eagerly await your further response.

---

> ### Author Response · Authors · 2023-11-17
> **Does our response address your concerns?**
>
> Dear Reviewer CYAM,
>
> Thank you for reviewing our work and helping us enhance it. In this revision, we've tried our best to **address any contradictory in the paper** and **compare the existing ULTR algorithms**. Please inform us if these improvements address your concerns. We're open for further discussion if necessary. Thank you for your time and consideration.
>
> Best regards,
>
> Authors

---

> ### Author Response · Authors · 2023-11-19
> **Discussion period will end soon and we're eagerly waiting for your feedback**
>
> Dear Reviewer CYAM,
>
> Hope you are well! ***The discussion is coming to an end***, and we're really excited to hear what you think about our response. We're just checking to see if our rebuttal improved the paper. It would be wonderful if you could tell us if we addressed your previous concerns. Thank you so much for your dedication to reviewing our paper.
>
> Best wishes,
>
> The Authors

---

### Official Review · Reviewer_tHmn · 2023-10-31

**Soundness:** 2 fair
**Presentation:** 3 good
**Contribution:** 3 good
**Rating:** 5
**Confidence:** 4

**Summary:**

This paper delves into the identifiability issue of the ranking model within the context of unbiased learning to rank (ULTR). Previous studies have established the model's unbiasedness, assuming perfect fits for both clicks and the observation model, inadvertently overlooking the identifiability challenge at its core. Motivated by this, the article investigates the conditions necessary to recover identifiability, primarily in the context of scale transformation. The authors formalize the identifiability challenge as a graph connectivity test problem. Based on it, they further propose two methods, namely node intervention and node merging, to tackle this problem for empirical applications.

**Strengths:**

1. The topic is interesting and important both theoretically and empirically;
2. The methods of graph connectivity are novel to me;
3. This paper is well-written.

**Weaknesses:**

1. Experiments are weak and not very convincing, since it has very little baseline (see Q1 below for more details);

2. Theorem 3 is straightforward and does not seem useful; the conditions required by Theorems 2 and 4 are stringent (see Q2 below for more details).

**Questions:**

I have two main concerns:

**Q1.** In the experiment, there are only two weak baselines. In addition, there are lack of details about the two baselines. However, this article mentions a lot of related works but does not include them as part of the baselines for comparison empirically. This is inconsistent with the requirements in this area of ULTR. Could you add some cutting-edge ULTR methods as baselines?

**Q2.** In terms of the theoretical results.

>(1)	Theorem 2 is a very simple case (… $x$ and $t$ are selected independently and uniformly …). Thus, it is not sufficient to write it as a Theorem; it might be more appropriate to write it as a Proposition.

> (2)	Theorem 3 is simple and straightforward (just by the central limit theorem) and does not seem useful. Could you clarify the purpose and use of Theorem 3? Also, it is not sufficient to write it as a Theorem; it might be more appropriate to write it as a Lemma.

> (3)	The condition required in Theorem 4, "A disconnected IG consists of two connected components G1 and G2" is strong and difficult to fulfill in practice.

**In fact, the soundness of Theorems 3 and 4 is very critical to this paper.** Here are the two main reasons: (a) The graph is always disconnected in practice, and will suffer from identifiability problems; (b) To recover the identifiability, we always need node intervention and node merging to recover the connected graph for empirical applications. Thus, the rationality of these two proposed algorithms (node intervention and node merging) becomes critical. Regrettably, Theorems 3 and 4 are slightly weak, which seriously undermines the contribution of this paper.

---

> ### Author Response · Authors · 2023-11-13
>
> Thank your for your valuable feedback. Below are our responses to the weaknesses and questions:
>
> > Experiments are weak and not very convincing, since it has very little baseline.
>
> We appreciate your feedback about the evaluation. **We've made changes to the paper, and now it includes results for DLA, Regression-EM, and Two-Tower**. Please check our general response for more details.
>
> > (1) Theorem 2 is a very simple case. Thus, it is not sufficient to write it as a Theorem; it might be more appropriate to write it as a Proposition.
>
> Thanks for pointing out the problem. This theorem is not critical for our proposed method and the experiments we conducted. Its purpose is to provide an intuition of how "increasing bias factors or decreasing data can potentially lead to unidentifiability" in a simple scenario. We've verified this in a more complex and practical setting in Section 5.2. **We appreciate your suggestion of renaming and we've renamed Theorem 2 to Corollary 1.**
>
> > (2) Theorem 3 is simple and straightforward (just by the central limit theorem) and does not seem useful. Could you clarify the purpose and use of Theorem 3? Also, it is not sufficient to write it as a Theorem; it might be more appropriate to write it as a Lemma.
>
> Thanks for pointing out this point. We kindly remind that this theorem is necessary in the proposed node intervention method: The purpose is to used to compute the specific form of variance, which serves as the cost function (Eq.3). The basic idea is that there are too many choices that the intervention targets can be selected, and in Theorem 3 we find that minimizing the variance helps facilitate the identifiability. This is why we've chosen variance as our cost function for choosing intervention target, and **we've highlighted this point in Theorem 3, which is now renamed Proposition 1.**
>
> > (3) The condition required in Theorem 4, "A disconnected IG consists of two connected components G1 and G2" is strong and difficult to fulfill in practice.
>
> Apologies for the confusion about the theorem name "error bound of merging". Its purpose is not to establish the error bound for node merging but to illustrate the error when combining two components, which forms the foundation for the below merging cost (Eq.7). **To address the error bound for node merging, we have created another corollary in Appendix B.5.**
>
> ---
>
> We hope the above responses and revisions could strengthen Theorem 3 and 4. Please confirm whether our explanations address your concerns. If you have any more questions or additional concerns, please feel free to ask, and we will make every effort to address them.

---

> ### Author Response · Authors · 2023-11-17
> **Does our response address your concerns?**
>
> Dear Reviewer tHmn,
>
> Thank you for reviewing our work and aiding in its improvement. In this revision, we've **compared the existing ULTR algorithms**, **clarified the motivation behind Theorem 3**, and **introduced an additional corollary to fortify Theorem 4 for increased reliability**. Please let us know if these efforts address your concerns. We remain open for further discussion if needed. Thank you for your time and consideration.
>
> Best regards,
>
> Authors

---

> ### Author Response · Authors · 2023-11-19
> **Discussion period will end soon and we're eagerly waiting for your feedback**
>
> Dear Reviewer tHmn,
>
> I hope you're doing well! ***The discussion period is ending soon***, and we're excited to know your thoughts on our response. We're just checking in to see if our rebuttal improved the paper. It would mean a lot if you could kindly inform us if we addressed your previous concerns properly. Thank you for your dedication to reviewing our paper.
>
> Best regards,
>
> The Authors

---

### Official Review · Reviewer_ggp4 · 2023-11-01

**Soundness:** 2 fair
**Presentation:** 3 good
**Contribution:** 2 fair
**Rating:** 6
**Confidence:** 3

**Summary:**

The paper investigates the problem of position bias in the task of Unbiased Learning to Rank. It first introduces the widespread concern of biased user logs and the consequent appriximation error in most of the commonly used ranking models. With a clear problem setup and definition of identifiability, the author states the conditions under which the true relevances can be extracted from click data. Particularly, the paper converts the identifiability problem into a graph connectivity problem. Based on the connectivity problem,  the authors further come up two new approaches to deal with “unidentifiable” datasets and optimize the ranking models using structures of the graph. Experiments are conducted to prove the validity of the graph conversion, the performance of new approaches, and the application to the real-world datasets.

**Strengths:**

S1: As one of its major contributions, this paper tansfers the identifiability of a ULTR task to the connectivity of a graph constructed based on the dataset. This equivalence is useful in that tasks related to a graph usually have more efficient computations, nicer properties, and more intuition-based understandings. The work also allows for more explainability in the field of ULTR and thus simplifies difficult questions.

S2: This paper proposes two novel methods, node intervention and node merging, to bridge the “unidentifiability gap” by utilizing the graph properties. These two methods are theoretically supported and empirically verified.

**Weaknesses:**

W1: Since the theory of this paper relies heavily on the examination hypothesis, the graph-equivalence idea is not generalizeable to other general hypotheses on the dataset.

W2: Since the methods are still propensity-based, there are a plethora of such ranking models. It would be fair game if the paper uses such methods as baselines to strengthen the validity of the proposed methods.

W3: When choosing the bias factors, we can choose either fewer factors, which makes the graph more likely to be connected, or more factors, which accounts for more bias but poses a more disconnected graph. It would be great if there is any discussion on the tradeoff and the corresponding performance of the two proposed methods. In addition, assume that each feature x and bias factor t are independently and uniformly chosen to construct the dataset D is nearly impossible in practice.

**Questions:**

Q1: In the real world, the dataset is mostly sparse and thus there might be a large number of connected components in IG. How much does the performance of the two methods deteriote with the increasing sparsity? Is there a systematic way to deal with that issue?

Q2: In the node merging method, the costs between nodes are computed based on the their deterministic features X_t. However, how is it guaranteed that the features reflect their true similarity? For example, we may use document rank as a bias factor when only considering the position bias. But it turns out that the user may notice the documents in the order of: the first several documents (since they’re most noticeable), the last several documents on this page (since users may scroll down), and then documents in the middle. In the more complex settings of several factors, it’s even less obvious which nodes are similar to each other. Is it possible to make the features not deterministic but rather learned throughout multi-task learning?

Q3: In the application of the method, the dataset is mostly online and continuously taking in new data points. How does the proposed method handle the updates efficiently? For example, if two nodes (bias factors) with similar features are already merged but the new datapoints from the user creates an edge between them, is there a way to efficiently deal with this?

---

> ### Author Response · Authors · 2023-11-13
>
> Thank your for your valuable feedback. Below are our responses to the weaknesses and questions:
>
> > W1: Since the theory of this paper relies heavily on the examination hypothesis, the graph-equivalence idea is not generalizeable to other general hypotheses on the dataset.
>
> We agree that this paper is based on the examination hypothesis. We opted to focus on it because it's the most widely used generation process (see Appendix A). However, we kindly remind that it does not mean the graph idea is not applicable to other hypothesis. **Actually, to generalize this idea to other hypothesis, only small adjustments in Definition 1 and Theorem 1 are required while maintaining the core structure of IG**. We leave it as future work since it is more complicated and falls beyond the scope of this work.
>
> > W2: Since the methods are still propensity-based, there are a plethora of such ranking models. It would be fair game if the paper uses such methods as baselines to strengthen the validity of the proposed methods.
>
> We appreciate your feedback about the evaluation. **We've made changes to the paper, and now it includes results for DLA, Regression-EM, and Two-Tower**. Please check our general response for more details.
>
> > W3 (1): It would be great if there is any discussion on the trade-off between bias vs connected and the corresponding performance of the two proposed methods.
>
> We report the MCC performance of (1) only considering one bias factor; (2) considering all bias factors; and (3)(4) our proposed methods in the K=2 simulation dataset:
>
> |ULTR method|DLA|Regression-EM|Two-Tower|
> |-|-|-|-|
> |(1) one bias factor|0.641|0.641|0.641|
> |(2) all bias factors|0.738|0.634|0.892|
> |(3) node intervention|1.000|0.982|1.000|
> |(4) node merging|0.987|0.975|0.987|
>
> The trade-off depends on the specific ULTR algorithms: Regression-EM faces more trouble with unidentifiability, whereas DLA and Two-Tower have more issues with bias. This happens because when there's no assurance of identifiability, different ULTR algorithms might end up with various poor model parameters. But when our methods ensure identifiability, they all converge to a shared set of good settings that recover the relevance accurately.
>
> > W3 (2): In addition, assume that each feature x and bias factor t are independently and uniformly chosen to construct the dataset D is nearly impossible in practice.
>
> Thanks for pointing it out. Note that this theorem is not critical for our proposed method and the experiments we conducted. Its purpose is to provide an intuition of how "increasing bias factors or decreasing data can potentially lead to unidentifiability" in a simple scenario. We've verified this in a more complex and practical setting in Section 5.2. **To reduce its significance, we've renamed Theorem 2 to Corollary 1.**
>
> > Q1: How much does the performance of the two methods deteriote with the increasing sparsity of IG?
>
> We presents the `nDCG@10 / MCC` results with the increasing numbers of connected components (K) in the fully-simulation datasets as follows:
>
> ||K=1|K=2|K=3|K=4|
> |-|-|-|-|-|
> |No debias|0.863 / 0.657|0.858 / 0.641|0.859 / 0.638|0.870 / 0.655|
> |DLA|1.000 / 1.000|0.900 / 0.738|0.889 / 0.700|0.876 / 0.660|
> |+ Node intervention |-|1.000 / 1.000 |1.000 / 1.000|1.000 / 1.000|
> |+ Node merging |-|1.000 / 0.987|1.000 / 0.978|0.947 / 0.835|
>
> We can observe that the performance of DLA and Node merging drops as the $K$ increases, while Node intervention always recovers the relevance accurately. Overall, our proposed methods significantly improve the initial results.
>
> > Is there a systematic way to deal with that issue?
>
> If doing intervention experiments is allowed, node intervention is always perfered. Otherwise, node merging is also acceptable since it can siginificantly reduce the unidentifiability issue.
>
> > Q2: Is it possible to make the features not deterministic but rather learned throughout multi-task learning?
>
> This is an interesting question. This depends on prior knowledge, often needing practitioners to manually define click models based on their experience to represent this bias similarity. While it's beyond this paper's scope, we speculate that automatically learning this similarity might not be viable. This is because unidentifiability arises from incomplete datasets and limited information, which challenges learning the actual similarity.
>
> > Q3: In the application of the method, the dataset is mostly online and continuously taking in new data points. How does the proposed method handle the updates efficiently?
>
> We can maintain a set recording all the merging pairs. When new data arrives, we can check if it's already in the merging set. If it is, we can separate the two nodes and handle them individually in the upcoming training.
>
> ---
>
> Please confirm whether our explanations address your concerns. If you have any more questions or additional concerns, please feel free to ask, and we will make every effort to address them.

---

> ### Author Response · Authors · 2023-11-17
> **Does our response address your concerns?**
>
> Dear reviewer ggp4,
>
> Thanks for reviewing our work and joining in to help make the paper better. We've tried our best to **do the experiments you asked for and answer your questions**. Please let us know if our responses have addressed your concerns. We're ready to talk more if needed. Thanks for your time and thinking about this.
>
> Best,
>
> Authors.

---

> ### Author Response · Authors · 2023-11-19
> **Discussion period will end soon and we're eagerly waiting for your feedback.**
>
> Dear Reviewer ggp4,
>
> I hope you're doing well! ***The discussion period is ending soon***, and we're really excited to hear your thoughts on our response. We just want to check if our rebuttal helped to improve the paper. It would be greatly appreciated if you could kindly tell us if we addressed your previous concerns properly. Thank you once more for your commitment to reviewing our paper.
>
> Best wishes,
>
> The Authors

---

> > ### Comment · Reviewer_ggp4 · 2023-11-22
> >
> > Dear authors,
> >
> > Thank you for your responses. On W2, the three extra baseline models provides more solid practical support for the proposed novel techniques in the paper. On W3(1), it’s interesting to see how different models deal with the trade-off between more bias factors and more connected IG. It’s also good to see that the proposed techniques handle the issues properly.  On Q1, the follow-up experiment on disconnectedness verifies the intuitions of the phenomenon and strengthens the validity of the proposed methods. Therefore, I would like to raise the rating to 6.
> >
> > Best,
> > Reviewer ggp4

---

> > > ### Author Response · Authors · 2023-11-22
> > > **Thanks again for your time and efforts**
> > >
> > > Dear Reviewer ggp4,
> > >
> > > We are delighted to see that our follow-up experiments effectively addresses your concerns. We really appreciate your efforts in reviewing our work and response, and updating the score. Your invaluable feedback has been instrumental in enhancing the quality of our work.
> > >
> > > Best wishes,
> > >
> > > The Authors

---

### Official Review · Reviewer_2RKu · 2023-11-02

**Soundness:** 3 good
**Presentation:** 3 good
**Contribution:** 3 good
**Rating:** 5
**Confidence:** 4

**Summary:**

In this paper, the authors explore if or when the true relevance can be recovered from click data. Overall, it is a solid paper. My concern lies in whether and how can this approach apply to the existing unbiased learning-to-rank framework that developed from the examination hypothesis. Or, how the proposed framework incorporates current ranking models. Also, it would be better to compare this work against more recently proposed existing unbiased learning-to-rank algorithms. Overall, I give a weak rejection. If the authors would clarify the above concerns, I will be happy to raise my score.

**Strengths:**

1. It is an interesting and important topic to investigate when the true relevance can be recovered from click data.
2. I like the theoretical analysis in this paper (i.e., Sections 3 and 4).
3. They have conducted experiments on Yahoo! and Istella-S datasets, verifying the performance of the proposed model.

**Weaknesses:**

1. It lacks recently proposed methods as the baselines.
2. It would be interesting to discuss the difference between the proposed method and the existing approaches based on the examination hypothesis.
3. For an unbiased learning-to-rank algorithm, there is always a ranking algorithm base. It is not clear how the proposed model incorporates the existing ranking models.

**Questions:**

It is essential to evaluate if or when the true relevance can be recovered from click data. I like this idea very much. However, in the context of unbiased learning-to-rank, there should be a ranking function (often referred to as biased), and then the core goal of unbiased learning-to-rank is to build a debiasing method that can be incorporated into the biased ranking models. After reading this paper multiple times, I consider that it is not clear how this approach can be applied to existing ranking models. Also, in the experiment part, the authors only compare no debias and a simple examine hypothesis method. I highly recommend the authors compare and discuss the connections to existing unbiased learning-to-rank algorithms such as “Unbiased Learning to Rank with Unbiased Propensity Estimation” and “An Unbiased Pairwise Learning-to-Rank Algorithm”. Therefore, I would like to give a weak rejection.  If the authors would clarify the above concerns, I will be happy to raise my score.

---

> ### Author Response · Authors · 2023-11-13
>
> Thank your for your valuable feedback. We are glad that you enjoy the idea in our work. Below are our responses to the weaknesses and questions:
>
> > W1 & Q: It lacks recently proposed methods as the baselines.
>
> We appreciate your feedback about the evaluation. **We've made changes to the paper, and now it includes results for DLA, Regression-EM, and Two-Tower**. Please check our general response for more details. Our initial submission is linked with DLA, the paper you first shared, as mentioned in Appendix E.3. As for the second paper you shared (known as Pairwise-Debiasing), we didn't compare with it for specific reasons:
>
> - It is not a theoretically sound method, which is proved in [1] (Section 7.1) and [2] (Section 4.1.4).
>
> - It's not based on examination hypothesis, which is our framework developed on.
>
> [1] Reaching the End of Unbiasedness: Uncovering Implicit Limitations of Click-Based Learning to Rank.
>
> [2] Unbiased Learning to Rank: Online or Offline?
>
> > W2. It would be interesting to discuss the difference between the proposed method and the existing approaches based on the examination hypothesis.
> >
> > W3. For an unbiased learning-to-rank algorithm, there is always a ranking algorithm base. It is not clear how the proposed model incorporates the existing ranking models.
> >
> > Q: I consider that it is not clear how this approach can be applied to existing ranking models.
>
> **Our proposed algorithms work with the dataset**: First, we use algorithm 1 to assess if the dataset is identifiable. If it's not, we use algorithm 2 and 3 to adjust the dataset and ensure that its IG is connected. Once that's done, **we apply existing ULTR algorithms and train ranking models using this processed dataset**. This means our method is compatible with various models. We've explained this in Section 5.1 (in initial submission) and explicitly mentioned it in the Introduction (in the revised paper).
>
> ---
>
> Please confirm whether our explanations address your concerns. If you have any more questions or additional concerns, please feel free to ask, and we will make every effort to address them.

---

> ### Author Response · Authors · 2023-11-15
> **Does our response address your concerns?**
>
> Dear reviewer 2RKu,
>
> Thanks for reviewing our work and participating in the rebuttal process to improve the paper. We've **compared the existing ULTR algorithms in the revision** and explained **how our methods can be applied to existing methods**. Please let us know if our responses have addressed your concerns. We're open to further discussions if necessary. Appreciate your time and consideration.
>
> Best,
>
> Authors.

---

> ### Author Response · Authors · 2023-11-19
> **Discussion period will end soon and we're eagerly waiting for your feedback.**
>
> Dear Reviewer 2RKu,
>
> Hope you are well! ***The discussion time is wrapping up soon***, and we're really eager to hear your thoughts on our response. We're just checking in to see if our rebuttal helped to make the paper better. It would mean a lot if you could kindly let us know whether we've addressed your earlier concerns properly. Thanks again for your dedication to reviewing our paper.
>
> Warm regards,
>
> The Authors.

---

### Author Response · Authors · 2023-11-13
**General response by Authors**

We thank the reviewers for their insightful comments and valuable feedbacks. We are delighted that all reviewers agree that our proposed identifiability framework is novel, interesting and important for the ULTR research. They agree that the paper is well-written (R tHmn and R CYAM) with interesting theoretical analysis (R 2rku and R tHmn). We appreciate the reviewers valuable suggestions and will incorporate all feedback into the final version of our work.

In response to the reviewers' concerns, we have made revisions to the submission, which are indicated in red. Herein, we provide a summary of the key changes made in response to their feedback.

- One common concern (R ggp4, R tHmn, R CYAM and R 2rku) is whether our proposed methods work well with state-of-the-art baselines. In the revision, **we show how our methods perform when used with three different types of ULTR optimization algorithms: DLA [1], Regression-EM [2] and Two-Tower [3] in Table 1**. The results we initially shared were combined with DLA, as explained in Appendix E.3. We highlight that many recent strong baselines are variants of these three types of ULTR algorithms with some difference in bias factor considerations, like [4] using *outliers* with Regression-EM, and [5] using *MTypes, Serphs, and Slipoff* with DLA. Here are some of the results we're reporting:

|                     | MCC   | nDCG@10 |
| ------------------- | ----- | ------- |
| No debias           | 0.641 | 0.858   |
| DLA                 | 0.738 | 0.900   |
| + Node intervention | 1.000 | 1.000   |
| + Node merging      | 0.987 | 1.000   |
| Regression-EM       | 0.634 | 0.868   |
| + Node intervention | 0.982 | 0.999   |
| + Node merging      | 0.975 | 1.000   |
| Two-Tower           | 0.892 | 0.961   |
| + Node intervention | 1.000 | 1.000   |
| + Node merging      | 0.987 | 1.000   |

Without a guarantee of identifiability, various ULTR algorithms can end up converging to poor relevance recovery. However, when our proposed methods ensure identifiability, they converge to accurately recover the relevance. This demonstrates the model-agnostic nature of our methods.

- Another concern is the strong assumptions in our theoretical results (R tHmn, R ggp4 and R CYAM):
  - In Theorem 2, note that this theorem is not critical for our proposed method and the experiments we conducted. Its purpose is to provide an intuition of how "increasing bias factors or decreasing data can potentially lead to unidentifiability" in a simple scenario. We've verified this in a more complex and practical setting in Section 5.2. **To reduce its significance, we've renamed Theorem 2 to Corollary 1.**
  - In Theorem 3, we've discovered that the assumption "N is sufficiently large" is not necessary for our proposed method, as it's only used for applying the central limit theorem. **We have removed this unnecessary assumption and restructured the theorem to present only the variance term, which is all our method requires.**
  - For Theorem 4, its purpose is not to derive the error bound for node merging (apologies for the misleading name). Instead, it is used to illustrate the error when merging two components, providing a strong foundation for the below merging cost. **We have created another corollary in Appendix B.5 to give the true error bound for node merging.**

We hope the above revision could address the concerns in experiments and theorems. In the following, we will address the concern of each Reviewer individually.

[1] Unbiased learning to rank with unbiased propensity estimation.

[2] Position bias estimation for unbiased learning to rank in personal search.

[3] Pal: a position-bias aware learning framework for ctr prediction in live recommender systems.

[4] On the impact of outlier bias on user clicks.

[5] Multi-feature integration for perception-dependent examination-bias estimation.

---

### Author Response · Authors · 2023-11-23
**We would like to summarize our contribution and revisions, and kindly ask for a reevaluation**

Dear Reviewers and ACs,

As the discussion is ending soon in the next hours, we're grateful for the helpful comments and smart suggestions from all reviewers. **We would really appreciate it if the reviewers (R tHmn, R CYAM and R 2rku) could reevaluate the reviews considering our follow-up experiments and theoretical revisions**. In the following, we would like to summarize the contribution and revisions of this paper again.

**Contributions**:

- ***Core contribution***: we address the identifiability for ULTR from a new perspective of graph connectivity. This perspective is ``interesting``, ``important`` and ``novel`` (*R ggp4, R tHmn, R CYAM*), ``useful``, ``have nicer properties``, ``more intuition-based understandings``, ``more explainability`` and ``able to simplify difficult questions`` (*R ggp4*), and also ``lay the foundation for future work`` (*our response to R ggp4*).

- The proposed two methods to deal with unidentifiability issue based on this graph view are ``novel and effective`` (*R ggp4*), and ``easy to understand`` (*R CYAM*).

- The proposed methods are supported by both ``interesting theoretical analysis`` (*R 2rku, R tHmn*) and ``the extensive experiments`` (*R 2rku*). The paper is well-written (*R tHmn, R CYAM*).

**Weaknesses and revisions**:

- The common concern (*R ggp4, R tHmn, R CYAM and R 2rku*) is whether our proposed methods work well with state-of-the-art baselines. In the revision, we show how our methods perform when used with three different types of ULTR optimization algorithms: DLA [1], Regression-EM [2] and Two-Tower [3] in Table 1. **This follow-up experiment has been approved by R ggp4**, which ``provides more solid practical support``.

- Another concern is the strong assumptions in our theoretical results (*R tHmn, R ggp4 and R CYAM*). We make revisions to the theorems as follows.
    - Theorem 2: this theorem is not so significant and thus we **rename** it to Corollary 1. Also, we've verified its conclusion in a more complex and practical setting in the experiment.
    - Theorem 3: we **remove the unnecessary assumption** (*"N is sufficiently large"*) and restructure the theorem.
    - Theorem 4: we have **created another corollary** to give the true error bound for node merging.

- Besides, we also conducted a thorough experiment for tuning the feature numbers, confirming that increasing the dimension of $X$ can indeed lead to potential unidentifiability issues (*our response to R CYAM*). We also tried our best to address any contradictory in the paper (*our response to R CYAM*).

We would again like to thank all reviewers, and we hope that our changes adequately address the concerns.

Sincerely,

The authors

---

### Meta-Review · Area_Chair_V1BY · 2023-12-06

**Metareview:**

The paper studies the identifiability problem of unbiased learning to rank (ULTR) - given a dataset of implicit feedback, whether the true relevance can be identified or not. The authors formulate the task as a graph connectivity test problem, and propose two solutions, namely node intervention and node merging. Theoretical analysis and empirical results are presented to support the algorithms.

The reviewers all appreciate the novelty perspective of the identifiability problem of ULTR. However, reviewers have shared concerns regarding (1) insufficient/weak baselines, (2) assumptions of the theory are strong, (3) motivation of the setting and theorems should be clarified. The author provided helpful responses including additional experiments and revision of the theoretical results, but did not fully resolve the reviewers' concerns.

**Justification For Why Not Higher Score:**

Three out of four reviewers recommend rejection. The AC agrees with this decision. The substantial changes during rebuttal is helpful but require the paper to go through a full review cycle.

**Justification For Why Not Lower Score:**

N/A

---

### Decision · Program_Chairs · 2024-01-16

Reject